# Disciplined Saddle Programming

**Philipp Schiele**                                                                    *philipp.schiele@stat.uni-muenchen.de*
*Department of Statistics*
*Ludwig-Maximilians-Universität München*

**Eric Luxenberg**                                                                                  *ericlux@stanford.edu*
*Department of Electrical Engineering*
*Stanford University*

**Stephen Boyd**                                                                                      *boyd@stanford.edu*
*Department of Electrical Engineering*
*Stanford University*

**Reviewed on OpenReview:** *https://openreview.net/forum?id=KhMLfEIoUm*

## Abstract

We consider convex-concave saddle point problems, and more generally convex optimization problems we refer to as *saddle problems*, which include the partial supremum or infimum of convex-concave saddle functions. Saddle problems arise in a wide range of applications, including game theory, machine learning, and finance. It is well known that a saddle problem can be reduced to a single convex optimization problem by dualizing either the convex (min) or concave (max) objectives, reducing a min-max problem into a min-min (or max-max) problem. Carrying out this conversion by hand can be tedious and error prone. In this paper we introduce *disciplined saddle programming* (DSP), a domain specific language (DSL) for specifying saddle problems, for which the dualizing trick can be automated. The language and methods are based on recent work by Juditsky & Nemirovski (2022), who developed the idea of conic-representable saddle point programs, and showed how to carry out the required dualization automatically using conic duality. Juditsky and Nemirovski's conic representation of saddle problems extends Nesterov and Nemirovski's earlier development of conic representable convex problems; DSP can be thought of as extending disciplined convex programming (DCP) to saddle problems. Just as DCP makes it easy for users to formulate and solve complex convex problems, DSP allows users to easily formulate and solve saddle problems. Our method is implemented in an open-source package, also called DSP.

## 1 Introduction

We consider saddle problems, by which we mean convex-concave saddle point problems or, more generally, convex optimization problems that include the partial supremum or infimum of convex-concave saddle functions. Saddle problems arise in various fields such as game theory, robust and minimax optimization, machine learning, and finance.

While there are algorithms specifically designed to solve some types of saddle point or minimax problems, another approach is to convert them into standard convex optimization problems using a trick based on duality that can be traced back to at least the 1920s. The idea is to express the infima or suprema that appear in the saddle problem via their duals, which converts them to suprema or infima, respectively. Roughly speaking, this turns a min-max problem into a min-min (or max-max) problem, which can then be solved by standard methods. Specific cases of this trick are well known; the classical example is converting a matrix game, a specific saddle point problem, into a linear program (LP) (Morgenstern & Von Neumann,

1953). While the dualizing trick has been known and used for almost 100 years, it has always been done by hand, for specific problems. It can only be carried out by those who have a working knowledge of duality in convex optimization, and are aware of the trick.

In this paper we propose an automated method for carrying out the dualizing trick. Our method is based on the theory of conic representation of saddle point problems, developed recently by Juditsky & Nemirovski (2022). Based on this development, we have designed a domain specific language (DSL) for describing saddle problems, which we refer to as disciplined saddle programming (DSP). When a problem description complies with the syntax rules, *i.e.*, is DSP-compliant, it is easy to verify that it is a valid saddle problem, and more importantly, automatically carry out the dualizing trick. We have implemented the DSL in an open source software package, also called DSP, which works with CVXPY (Diamond & Boyd, 2016), a DSL for specifying and solving convex optimization problems. DSP makes it easy to specify and solve saddle problems, without any expertise in (or even knowledge of) convex duality. Even for those with the required expertise to carry out the dualizing trick by hand, DSP is less tedious and error prone.

DSP is *disciplined*, meaning it is based on a small number of syntax rules that, if followed, guarantee that the specified problem is a valid saddle problem. It is analogous to disciplined convex programming (DCP) (Grant et al., 2006), which is a DSL for specifying convex optimization problems. When a problem specification follows these syntax rules, *i.e.*, is DCP-compliant, it is a valid convex optimization problem, and more importantly can be automatically converted to an equivalent cone program, and then solved. As a practical matter, DCP allows a large number of users to specify and solve even complex convex optimization problems, with no knowledge of the reduction to cone form. Indeed, most DCP users are blissfully unaware of how their problems are solved, *i.e.*, a reduction to cone form. DCP was based on the theory of conic representations of convex functions and problems, pioneered by Nesterov & Nemirovski (1992). Widely used implementations of DCP include CVXPY (Diamond & Boyd, 2016), Convex.jl Udell et al. (2014), CVXR (Fu et al., 2020), YALMIP (Lofberg, 2004), and CVX (Grant & Boyd, 2014). Like DCP did for convex problems, DSP makes it easy to specify and solve saddle problems, with most users unaware of the dualization trick and reduction used to solve their problems.

## 1.1 Previous and related work

**Saddle problems.** Studying saddle problems is a long-standing area of research, resulting in many theoretical insights, numerous algorithms for specific classes of problems, and a large number of applications.

Saddle problems are often studied in the context of minimax or maximin optimization (Dem'yanov & Malozemov, 1990; Du & Pardalos, 1995), which, while dating back to the 1920s and the work of von Neumann and Morgenstern on game theory (Morgenstern & Von Neumann, 1953), continue to be active areas of research, with many recent advancements for example in machine learning (Goodfellow et al., 2014). A variety of methods have been developed for solving saddle point problems, including interior point methods (Halldórsson & Tütüncü, 2003; Nemirovski, 1999), first-order methods (Korpelevich, 1976; Nemirovski, 2004; Nesterov, 2007; Nedić & Ozdaglar, 2009; Chen et al., 2013), and second-order methods (Nesterov & Polyak, 2006; Nesterov, 2008), where many of these methods are specialized to specific classes of saddle problems. Depending on the class of saddle problem, the methods differ in convergence rate. For example, for the subset of smooth minimax problems, an overview of rates for different curvature assumptions is given in Thekumparampil et al. (2019). Due to their close relation to Lagrange duality, saddle problems are commonly studied in the context of convex analysis (see, for example, Boyd & Vandenberghe (2004, §5.4), Rockafellar (1970, §33–37), Rockafellar & Wets (2009, §11.J), Borwein & Lewis (2006, §4.3)), with an analysis via monotone operators given in Ryu & Yin (2022).

The practical usefulness of saddle programming in many applications is also increasingly well known. Many applications of saddle programming are robust optimization problems (Bertsimas et al., 2011; Ben-Tal et al., 2009). For example, in statistics, distributionally robust models can be used when the true distribution of the data generating process is not known (Dou & Anitescu, 2019). Another common area of application is in finance, with Cornuéjols et al. (2018, §19.3–4) describing a range of financial applications that can be characterized as saddle problems. Similarly, Boyd et al. (2017); Goldfarb & Iyengar (2003); Lobo & Boyd (2000) describe variations of the classical portfolio optimization problem as saddle problems.

**Disciplined convex programming.** DCP is a grammar for constructing optimization problems that are provably convex, meaning that they can be solved globally, efficiently and accurately. It is based on the rule that the convexity of a function $f$ is preserved under composition if all inner expressions in arguments where $f$ is nondecreasing are convex, and all expressions where $f$ is nonincreasing are concave, and all other expressions are affine. A detailed description of the composition rule is given in Boyd & Vandenberghe (2004, §3.2.4). Using this rule, functions can be composed from a small set of primitives, called atoms, where each atom has known curvature, sign, and monotonicity. Every function that can be constructed from these atoms according to the composition rule is convex, but the converse is not true. The DCP framework has been implemented in many programming languages, including MATLAB (Grant & Boyd, 2014; Lofberg, 2004), Python (Diamond & Boyd, 2016), R (Fu et al., 2020), and Julia (Udell et al., 2014), and is used by researchers and practitioners in a wide range of fields.

**Well-structured convex-concave saddle point problems.** As mentioned earlier, disciplined saddle programming is based on Juditsky and Nemirovski's recent work on well-structured convex-concave saddle point problems (Juditsky & Nemirovski, 2022).

### 1.2 Our contributions

We summarize our contributions as follows:

- We introduce disciplined saddle programming, a domain specific language for specifying and solving convex-concave saddle problems. To solve the saddle problems, automated dualization is applied to the conic representation of the problem. We extend the existing literature by deriving a procedure that returns both the convex and concave coordinates of the saddle point. This also guarantees that a valid saddle point was found without the need to check for technical conditions (such as compactness). These developments make the theory of conic representable saddle problems practically applicable for the first time.

- We specify and implement the first DSL that encodes sufficient conditions for conic representability of saddle problems. We develop an open-source Python package, also called DSP, providing a user-friendly interface for specifying and solving saddle problems. Using this implementation, we demonstrate the effectiveness of the framework by solving a variety of saddle problems from different application domains.

## 2 Saddle programming

### 2.1 Saddle functions

A *saddle function* (also referred to as a convex-concave saddle function) $f : \mathcal{X} \times \mathcal{Y} \to \mathbf{R}$ is one for which $f(\cdot, y)$ is convex for any fixed $y \in \mathcal{Y}$, and $f(x, \cdot)$ is concave for any fixed $x \in \mathcal{X}$. The argument domains $\mathcal{X} \subseteq \mathbf{R}^n$ and $\mathcal{Y} \subseteq \mathbf{R}^m$ must be nonempty closed convex. We refer to $x$ as the convex variable, and $y$ as the concave variable, of the saddle function $f$.

**Examples.**

- *Functions of $x$ or $y$ alone.* A convex function of $x$, or a concave function of $y$, are trivial examples of saddle functions.

- *Lagrangian of a convex optimization problem.* The convex optimization problem

$$\begin{array}{ll} \text{minimize} & f_0(x) \\ \text{subject to} & Ax = b, \quad f_i(x) \leq 0, \quad i = 1, \dots, m, \end{array}$$

  with variable $x \in \mathbf{R}^n$, where $f_0, \dots, f_m$ are convex and $A \in \mathbf{R}^{p \times n}$, has Lagrangian

$$L(x, \nu, \lambda) = f(x) + \nu^T (Ax - b) + \lambda_1 f_1(x) + \cdots + \lambda_m f_m(x),$$

for $\lambda \geq 0$ (elementwise). It is convex in $x$ and affine (and therefore also concave) in $y = (\nu, \lambda)$, so it is a saddle function with

$$\mathcal{X} = \bigcap_{i=0,\ldots,m} \mathbf{dom}\, f_i, \qquad \mathcal{Y} = \mathbf{R}^p \times \mathbf{R}^m_+,$$

- *Bi-affine function.* The function $f(x, y) = (Ax+b)^T(Cy+d)$, with $\mathcal{X} = \mathbf{R}^p$ and $\mathcal{Y} = \mathbf{R}^q$, is evidently a saddle function. The inner product $x^T y$ is a special case of a bi-affine function. For a bi-affine function, either variable can serve as the convex variable, with the other serving as the concave variable.

- *Convex-concave inner product.* The function $f(x, y) = F(x)^T G(y)$, where $F : \mathbf{R}^p \to \mathbf{R}^n$ is a nonnegative elementwise convex function and $G : \mathbf{R}^q \to \mathbf{R}^n$ is a nonnegative elementwise concave function.

- *Weighted $\ell_2$ norm.* The function

$$f(x, y) = \left( \sum_{i=1}^n y_i x_i^2 \right)^{1/2},$$

  with $\mathcal{X} = \mathbf{R}^n$ and $\mathcal{Y} = \mathbf{R}^n_+$, is a saddle function.

- *Weighted log-sum-exp.* The function

$$f(x, y) = \log \left( \sum_{i=1}^n y_i \exp x_i \right),$$

  with $\mathcal{X} = \mathbf{R}^n$ and $\mathcal{Y} = \mathbf{R}^n_+$, is a saddle function.

- *Weighted geometric mean.* The function $f(x, y) = \prod_{i=1}^n y_i^{x_i}$, with $\mathcal{X} = \mathbf{R}^n_+$ and $\mathcal{Y} = \mathbf{R}^n_+$, is a saddle function.

- *Quadratic form with quasi-semidefinite matrix.* The function

$$f(x, y) = \begin{bmatrix} x \\ y \end{bmatrix}^T \begin{bmatrix} P & S \\ S^T & Q \end{bmatrix} \begin{bmatrix} x \\ y \end{bmatrix},$$

  where the matrix is quasi-semidefinite, *i.e.*, $P \in \mathbf{S}^n_+$ (the set of symmetric positive semidefinite matrices) and $-Q \in \mathbf{S}^n_+$.

- *Quadratic form.* The function $f(x, Y) = x^T Y x$, with $\mathcal{X} = \mathbf{R}^n$ and $\mathcal{Y} = \mathbf{S}^n_+$ (the set of symmetric positive semidefinite $n \times n$ matrices), is a saddle function.

- As a more esoteric example, the function $f(x, Y) = x^T Y^{1/2} x$, with $\mathcal{X} = \mathbf{R}^n$ and $\mathcal{Y} = \mathbf{S}^n_+$, is a saddle function.

**Combination rules.** Saddle functions can be combined in several ways to yield saddle functions. For example the sum of two saddle functions is a saddle function, provided the domains have nonempty intersection. A saddle function scaled by a nonnegative scalar is a saddle function. Scaling a saddle function with a nonpositive scalar, and swapping its arguments, yields a saddle function: $g(x, y) = -f(y, x)$ is a saddle function provided $f$ is. Saddle functions are preserved by pre-composition of the convex and concave variables with an affine function, *i.e.*, if $f$ is a saddle function, so is $f(Ax + b, Cx + d)$. Indeed, the bi-affine function is just the inner product with an affine pre-composition for each of the convex and concave variables.

## 2.2 Saddle point problems

A *saddle point* $(x^\star, y^\star) \in \mathcal{X} \times \mathcal{Y}$ is any point that satisfies

$$f(x^\star, y) \leq f(x^\star, y^\star) \leq f(x, y^\star) \text{ for all } x \in \mathcal{X}, \ y \in \mathcal{Y}. \tag{1}$$

In other words, $x^\star$ minimizes $f(x, y^\star)$ over $x \in \mathcal{X}$, and $y^\star$ maximizes $f(x^\star, y)$ over $y \in \mathcal{Y}$. The basic *saddle point problem* is to find such a saddle point,

$$\text{find } x^\star, \ y^\star \text{ which satisfy equation 1.} \tag{2}$$

The value of the saddle point problem is $f(x^\star, y^\star)$.

Existence of a saddle point for a saddle function is guaranteed, provided some technical conditions hold. For example, Sion's theorem (Sion, 1958) guarantees the existence of a saddle point when $\mathcal{Y}$ is compact. There are many other cases.

**Examples.**

- *Matrix game.* In a matrix game, player one chooses $i \in \{1, \ldots, m\}$, and player two chooses $j \in \{1, \ldots, n\}$, resulting in player one paying player two the amount $C_{ij}$. Player one wants to minimize this payment, while player two wishes to maximize it. In a mixed strategy, player one makes choices at random, from probabilities given by $x$ and player two makes independent choices with probabilities given by $y$. The expected payment from player one to player two is then $f(x, y) = x^T C y$. With $\mathcal{X} = \{x \mid x \geq 0, \ \mathbf{1}^T x = 1\}$, and similarly for $\mathcal{Y}$, a saddle point corresponds to an equilibrium, where no player can improve her position by changing (mixed) strategy. The saddle point problem consists of finding a stable equilibrium, *i.e.*, an optimal mixed strategy for each player.

- *Lagrangian.* A saddle point of a Lagrangian of a convex optimization problem is a primal-dual optimal pair for the convex optimization problem.

## 2.3 Saddle extremum functions

Suppose $f$ is a saddle function. The function $G : \mathcal{X} \to \mathbf{R} \cup \{\infty\}$ defined by

$$G(x) = \sup_{y \in \mathcal{Y}} f(x, y), \quad x \in \mathcal{X}, \tag{3}$$

is called a *saddle max function.* Similarly, the function $H : \mathcal{Y} \to \mathbf{R} \cup \{-\infty\}$ defined by

$$H(x) = \inf_{x \in \mathcal{X}} f(x, y), \quad y \in \mathcal{Y}, \tag{4}$$

is called a *saddle min function.* Saddle max functions are convex, and saddle min functions are concave. We will use the term *saddle extremum* (SE) functions to refer to saddle max or saddle min functions. Which is meant is clear from context, *i.e.*, whether it is defined by minimization (infimum) or maximization (supremum), or its curvature (convex or concave). Note that in SE functions, we always maximize (or take supremum) over the concave variable, and minimize (or take infimum) over the convex variable. This means that evaluating $G(x)$ or $H(y)$ involves solving a convex optimization problem.

**Examples.**

- *Dual function.* Minimizing a Lagrangian $L(x, \nu, \lambda)$ over $x$ gives the dual function of the original convex optimization problem.

- Maximizing a Lagrangian $L(x, \nu, \lambda)$ over $y = (\nu, \lambda)$ gives the objective function restricted to the feasible set.

- *Conjugate of a convex function.* Suppose $f$ is convex. Then $g(x,y) = f(x) - x^T y$ is a saddle function, the Lagrangian of the problem of minimizing $f$ subject to $x = 0$. Its saddle min is the negative conjugate function: $\inf_x g(x,y) = -f^*(y)$.

- *Sum of $k$ largest entries.* Consider $f(x,y) = x^T y$, with $\mathcal{Y} = \{y \mid 0 \leq y \leq 1,\ \mathbf{1}^T y = k\}$. The associated saddle max function $G$ is the sum of the $k$ largest entries of $x$.

**Saddle points via SE functions.** A pair $(x^\star, y^\star)$ is a saddle point of a saddle function $f$ if and only if $x^\star$ minimizes the convex SE function $G$ in equation 3 over $x \in \mathcal{X}$, and $y^\star$ maximizes the concave SE function $H$ defined in equation 4 over $y \in \mathcal{Y}$. This means that we can find saddle points, *i.e.*, solve the saddle point problem equation 2, by solving the convex optimization problem

$$\begin{array}{ll} \text{minimize} & G(x) \\ \text{subject to} & x \in \mathcal{X}, \end{array} \tag{5}$$

with variable $x$, and the convex optimization problem

$$\begin{array}{ll} \text{maximize} & H(y) \\ \text{subject to} & y \in \mathcal{Y}, \end{array} \tag{6}$$

with variable $y$. The problem equation 5 is called a minimax problem, since we are minimizing a function defined as the maximum over another variable. The problem equation 6 is called a maximin problem.

While the minimax problem equation 5 and maximin problem equation 6 are convex, they cannot be directly solved by conventional methods, since the objectives themselves are defined by maximization and minimization, respectively. There are solution methods specifically designed for minimax and maximin problems (Lin et al., 2020; Mutapcic & Boyd, 2009), but as we will see minimax problems involving SE functions can be transformed to equivalent forms that can be directly solved using conventional methods.

## 2.4 Saddle problems

In this paper we consider convex optimization problems that include SE functions in the objective or constraints, which we refer to as *saddle problems.* The convex problems that solve the basic saddle point problem equation 5 and equation 6 are special cases, where the objective is an SE function. As another example consider the problem of minimizing a convex function $\phi$ subject to the convex SE constraint $H(y) \leq 0$, which can be expressed as

$$\begin{array}{ll} \text{minimize} & \phi(x) \\ \text{subject to} & f(x,y) \leq 0 \text{ for all } y \in \mathcal{Y}, \end{array} \tag{7}$$

with variable $x$. The constraint here is called a *semi-infinite constraint*, since (when $\mathcal{Y}$ is not a singleton) it can be thought of as an infinite collection of convex constraints, one for each $y \in \mathcal{Y}$ (Hettich & Kortanek, 1993).

Saddle problems include the minimax and maximin problems (that can be used to solve the saddle point problem), and semi-infinite problems that involve SE functions. There are many other examples of saddle problems, where SE functions can appear in expressions that define the objective and constraints.

**Robust cost LP.** As a more specific example of a saddle problem consider the linear program with robust cost,

$$\begin{array}{ll} \text{minimize} & \sup_{c \in \mathcal{C}} c^T x \\ \text{subject to} & Ax = b, \quad x \geq 0, \end{array} \tag{8}$$

with variable $x \in \mathbf{R}^n$, with $\mathcal{C} = \{c \mid Fc \leq g\}$. This is an LP with worst case cost over the polyhedron $\mathcal{C}$ (Bertsimas et al., 2011; Ben-Tal et al., 2009). This is a saddle problem with convex variable $x$, concave variable $y$, and an objective which is a saddle max function.

### 2.5 Solving saddle problems

**Special cases with tractable analytical expressions.** There are cases where an SE function can be worked out analytically. An example is the max of a linear function over a box,

$$\sup_{l \leq y \leq u} y^T x = (1/2)(u + l)^T x + (1/2)(u - l)^T |x|,$$

where the absolute value is elementwise. We will see other cases in our examples.

**Subgradient methods.** We can readily compute a subgradient of a saddle max function (or a supergradient of a saddle min function) at a given input, by simply maximizing over the concave variable (minimizing over the convex variable), which is itself a convex optimization problem, and then obtaining a subgradient (supergradient) at that maximizer (minimizer). We can then use any method to solve the saddle problem using these subgradients, *e.g.*, subgradient-type methods, ellipsoid method, or localization methods such as the analytic center cutting plane method. In Mutapcic & Boyd (2009) such an approach is used for general minimax problems.

**Methods for specific forms.** Many methods have been developed for finding saddle points of saddle functions with the special form

$$f(x, y) = x^T K y + \phi(x) + \psi(y),$$

where $\phi$ is convex, $\psi$ is concave, and $K$ is a matrix (Bredies & Sun, 2015; Condat, 2013; Chambolle & Pock, 2011; Nesterov, 2005a;b; Chambolle & Pock, 2016). Beyond this example, there are many other special forms of saddle functions, with different methods adapted to properties such as smoothness, separability, and strong-convex-strong-concavity.

### 2.6 Dual reduction

A well-known trick can be used to transform a saddle point problem into an equivalent problem that does not contain SE functions. This method of transforming an inner minimization is not new; it has been used since the 1950s when Von Neumann proved the minimax theorem using strong duality in his work with Morgenstern on game theory (Morgenstern & Von Neumann, 1953). Using this observation, he showed that the minimax problem of a two player game is equivalent to an LP. Duality allows us to express the convex (concave) SE function as an infimum (supremum), which facilitates the use of standard convex optimization. We think of this as a reduction to an equivalent problem that removes the SE functions from the objective and constraints.

**Robust cost LP.** We illustrate the dualization method for the robust cost LP equation 8. The key is to express the robust cost or saddle max function $\sup_{Fc \leq g} c^T x$ as an infimum. We first observe that this saddle max function is the optimal value of the LP

$$\begin{array}{ll} \text{maximize} & x^T c \\ \text{subject to} & Fc \leq g, \end{array}$$

with variable $c$. Its dual is

$$\begin{array}{ll} \text{minimize} & g^T \lambda \\ \text{subject to} & F^T \lambda = x, \quad \lambda \geq 0, \end{array}$$

with variable $\lambda$. With $\mathcal{C} = \{c \mid Fc \leq g\}$, and assuming $\mathcal{C}$ is nonempty, this dual problem has the same optimal value as the primal, *i.e.*,

$$\sup_{c \in \mathcal{C}} c^T x = \inf_{\lambda \geq 0, \ F^T \lambda = x} g^T \lambda$$

Substituting this into equation 8 we obtain the problem

$$\begin{array}{ll} \text{minimize} & g^T \lambda \\ \text{subject to} & Ax = b, \quad x \geq 0, \quad F^T \lambda = x, \quad \lambda \geq 0, \end{array} \tag{9}$$

with variables $x$ and $\lambda$. This simple LP is equivalent to the original robust LP equation 8, in the sense that if $(x^\star, \lambda^\star)$ is a solution of equation 9, then $x^\star$ is a solution of the robust LP equation 8.

We will see this dualization trick in a far more general setting in §4.

## 3    Applications

In this section we describe a few applications of saddle programming.

### 3.1    Robust bond portfolio construction

We describe here a simplified version of the problem described in much more detail in Luxenberg et al. (2022). Our goal is to construct a portfolio of $n$ bonds, giving by its holdings vector $h \in \mathbf{R}_+^n$, where $h_i$ is the number of bond $i$ held in the portfolio. Each bond produces a cash flow, *i.e.*, a sequence of payments to the portfolio holder, up to some period $T$. Let $c_{i,t}$ be the payment from bond $i$ in time period $t$. Let $y \in \mathbf{R}^T$ be the yield curve, which gives the time value of cash: A payment of one dollar at time $t$ is worth $\exp(-ty_t)$ current dollars, assuming continuously compounded returns. The bond portfolio value, which is the present value of the total cash flow, can be expressed as

$$V(h, y) = \sum_{i=1}^{n} \sum_{t=1}^{T} h_i c_{i,t} \exp(-ty_t).$$

This function is convex in the yields $y$ and concave (in fact, linear) in the holdings vector $h$.

Now suppose we do not know the yield curve, but instead have a convex set $\mathcal{Y}$ of possible values, with $y \in \mathcal{Y}$. The worst case value of the bond portfolio, over this set of possible yield curves, is

$$V^{\mathrm{wc}}(h) = \inf_{y \in \mathcal{Y}} V(h, y).$$

We recognize this as a saddle min function. (In this application, $y$ is the convex variable of the saddle function $V$, whereas elsewhere in this paper we use $y$ to denote the concave variable.)

We consider a robust bond portfolio construction problem of the form

$$\begin{array}{ll} \text{minimize} & \phi(h) \\ \text{subject to} & h \in \mathcal{H}, \quad V^{\mathrm{wc}}(h) \geq V^{\mathrm{lim}}, \end{array} \tag{10}$$

where $\phi$ is a convex objective, typically a measure of return and risk, $\mathcal{H}$ is a convex set of portfolio constraints (for example, imposing $h \geq 0$ and a total budget), and $V^{\mathrm{lim}}$ is a specified limit on worst case value of the portfolio over the yield curve set $\mathcal{Y}$, which has a saddle min as a constraint.

For some simple choices of $\mathcal{Y}$ the worst case value can be found analytically. One example is when $\mathcal{Y}$ has a maximum element. In this special case, the maximum element is the minimizer of the value over $\mathcal{Y}$ (since $V$ is a monotone decreasing function of $y$). For other cases, however, we need to solve the saddle problem equation 10.

### 3.2    Model fitting robust to data weights

We wish to fit a model parametrized by $\theta \in \Theta \subseteq \mathbf{R}^n$ to $m$ observed data points. We do this by minimizing a weighted loss over the observed data, plus a regularizer,

$$\sum_{i=1}^{m} w_i \ell_i(\theta) + r(\theta),$$

where $\ell_i$ is the convex loss function for observed data point $i$, $r$ is a convex regularizer function, and the weights $w_i$ are nonnegative. The weights can be used to adjust a data sample that was not representative,

as in Barratt et al. (2021), or to ignore some of the data points (by taking $w_i = 0$), as in Broderick et al. (2020). Evidently the weighted loss is a saddle function, with convex variable $\theta$ and concave variable $w$.

We consider the case when the weights are unknown, but lie in a convex set, $w \in \mathcal{W}$. The robust fitting problem is to choose $\theta$ to minimize the worst case loss over the set of possible weights, plus the regularizer,

$$\max_{w \in \mathcal{W}} \sum_{i=1}^{m} w_i \ell_i(\theta) + r(\theta).$$

We recognize the first term, *i.e.*, the worst case loss over the set of possible weights, as a saddle max function.

For some simple choices of $\mathcal{W}$ the worst case loss can be expressed analytically. For example with

$$\mathcal{W} = \{w \mid 0 \leq w \leq 1, \; \mathbf{1}^T w = k\},$$

(with $k \in [0, n]$), the worst case loss is given by

$$\max_{w \in \mathcal{W}} \sum_{i=1}^{m} w_i \ell_i(\theta) = \phi(\ell_1, \ldots, \ell_m),$$

where $\phi$ is the sum-of-$k$-largest entries (Boyd & Vandenberghe, 2004, §3.2.3). (Our choice of symbol $k$ suggests that $k$ is an integer, but it need not be.) In this case we judge the model parameter $\theta$ by its worst loss on any subset of $k$ of data points. Put another way, we judge $\theta$ by dropping the $m - k$ data points on which it does best (*i.e.*, has the smallest loss) (Broderick et al., 2020).

CVXPY directly supports the sum-of-$k$-largest function, so the robust fitting problem can be formulated and solved without using DSP. To support this function, CVXPY carries out a transformation very similar to the one that DSP does. The difference is that the transformation in CVXPY is specific to this one function, whereas the one carried out in DSP is general, and would work for other convex weight sets. One such case would be to constrain the Wasserstein distance of the weights to a nominal distribution.

### 3.3 Robust production problem with worst case prices

We consider the choice of a vector of quantities $q \in \mathcal{Q} \subseteq \mathbf{R}^n$. Positive entries indicate goods we buy, and negative quantities are goods we sell. The set of possible quantities $\mathcal{Q}$ is our production set, which is convex. In addition, we have a manufacturing cost associated with the choice $q$, given by $\phi(q)$, where $\phi$ is a convex function. The total cost is the manufacturing cost plus the cost of goods (which includes revenue), $\phi(q) + p^T q$, where $p \in \mathbf{R}^n$ is vector of prices.

We consider the situation when we do not know the prices, but we have a convex set they lie in, $p \in \mathcal{P}$. The worst case cost of the goods is $\max_{p \in \mathcal{P}} p^T q$. The robust production problem is

$$
\begin{array}{ll}
\text{minimize} & \phi(q) + \max_{p \in \mathcal{P}} p^T q \\
\text{subject to} & q \in \mathcal{Q},
\end{array}
\tag{11}
$$

with variable $q$. Here too we can work out analytical expressions for simple choices of $\mathcal{P}$, such as a range for each component, in which case the worst case price is the upper limit for goods we buy, and the lower limit for goods we sell. In other cases, we solve the saddle problem equation 11.

### 3.4 Robust Markowitz portfolio construction

Markowitz portfolio construction (Markowitz, 1952) chooses a set of weights (the fraction of the total portfolio value held in each asset) by solving the convex problem

$$
\begin{array}{ll}
\text{maximize} & \mu^T w - \gamma w^T \Sigma w \\
\text{subject to} & \mathbf{1}^T w = 1, \quad w \in \mathcal{W},
\end{array}
$$

where the variable is the vector of portfolio weights $w \in \mathbf{R}^n$, $\mu \in \mathbf{R}^n$ is a forecast of the asset returns, $\gamma > 0$ is the risk aversion parameter, $\Sigma \in \mathbf{S}_{++}^n$ is a forecast of the asset return covariance matrix, and $\mathcal{W}$ is a convex set of feasible portfolios. The objective is called the risk adjusted (mean) return.

Markowitz portfolio construction is known to be fairly sensitive to the (forecasts) $\mu$ and $\Sigma$, which have to be chosen with some care; see, *e.g.*, Black & Litterman (1991). One approach is to specify a convex uncertainty set $\mathcal{U}$ that $(\mu, \Sigma)$ must lie in, and replace the objective with its worst case (smallest) value over this uncertainty set. This gives the robust Markowitz portfolio construction problem

$$
\begin{array}{ll}
\text{maximize} & \inf_{(\mu, \Sigma) \in \mathcal{U}} \left( \mu^T w - \gamma w^T \Sigma w \right) \\
\text{subject to} & \mathbf{1}^T w = 1, \quad w \in \mathcal{W},
\end{array}
$$

with variable $w$. This is described in, *e.g.*, Boyd et al. (2017); Goldfarb & Iyengar (2003); Lobo & Boyd (2000). We observe that this is directly a saddle problem, with a saddle min objective, *i.e.*, a maximin problem.

For some simple versions of the problem we can work out the saddle min function explicitly. One example, given in Boyd et al. (2017), uses $\mathcal{U} = \mathcal{M} \times \mathcal{S}$, where

$$
\begin{array}{rcl}
\mathcal{M} & = & \{ \mu + \delta \mid |\delta_i| \leq \rho_i, \ i = 1, \ldots, n \}, \\
\mathcal{S} & = & \{ \Sigma + \Delta \mid \Sigma + \Delta \succeq 0, \ |\Delta_{ij}| \leq \eta (\Sigma_{ii} \Sigma_{jj})^{1/2}, \ i, j = 1, \ldots, n \},
\end{array}
$$

where $\rho > 0$ is a vector of uncertainties in the forecast returns, and $\eta \in (0, 1)$ is a parameter that scales the perturbation to the forecast covariance matrix. (We interpret $\delta$ and $\Delta$ as perturbations of the nominal mean and covariance $\mu$ and $\Sigma$, respectively.) We can express the worst case risk adjusted return analytically as

$$
\inf_{(\mu, \Sigma) \in \mathcal{U}} \left( \mu^T w - \gamma w^T \Sigma w \right) = \mu^T w - \gamma w^T \Sigma w - \rho^T |w| - \gamma \eta \left( \sum_{i=1}^{n} \Sigma_{ii}^{1/2} |w_i| \right)^2.
$$

The first two terms are the nominal risk adjusted return; the last two terms (which are nonpositive) represent the cost of uncertainty.

## 4 Disciplined saddle programming

### 4.1 Saddle function calculus

We use the notation $\phi(x, y) : \mathcal{X} \times \mathcal{Y} \subseteq \mathbf{R}^{n \times m} \to \mathbf{R}$ to denote a saddle function with concave variables $x$ and convex variables $y$. The set of operations that, when performed on saddle functions, preserves the saddle property are called the *saddle function calculus*. The calculus is quite simple, and consists of the following operations:

1. *Conic combination of saddle functions.* Let $\phi_i(x_i, y_i)$, $i = 1, \ldots, k$ be saddle functions. Let $\theta_i \geq 0$ for each $i$. Then the conic combination, $\phi(x, y) = \sum_{i=1}^{k} \theta_i \phi_i(x_i, y_i)$, is a saddle function.

2. *Affine precomposition of saddle functions.* Let $\phi(x, y)$ be a saddle function, with $x \in \mathbf{R}^n$ and $y \in \mathbf{R}^m$. Let $A \in \mathbf{R}^{n \times q}$, $b \in \mathbf{R}^n$, $C \in \mathbf{R}^{m \times p}$, and $d \in \mathbf{R}^m$. Then, with $u \in \mathbf{R}^q$ and $v \in \mathbf{R}^p$, the affine precomposition, $\phi(Au + b, Cv + d)$, is a saddle function.

3. *Precomposition of saddle functions.* Let $\phi(x, y) : \mathcal{X} \times \mathcal{Y} \subseteq \mathbf{R}^{n \times m} \to \mathbf{R}$ be a saddle function, with $x \in \mathbf{R}^n$ and $y \in \mathbf{R}^m$. The precomposition with a function $f : \mathbf{R}^p \to \mathbf{R}^n$, $\phi(f(u), y)$, is a saddle function if for each $i = 1, \ldots, n$ one of the following holds:

   - $f_i(u)$ is convex and $\phi$ is nondecreasing in $x_i$ for all $y \in \mathcal{Y}$ and all $x \in \mathcal{X}$.
   - $f_i(u)$ is concave and $\phi$ is nonincreasing in $x_i$ for all $y \in \mathcal{Y}$ and all $x \in \mathcal{X}$.

   Similarly, the precomposition with a function $g : \mathbf{R}^q \to \mathbf{R}^m$, $\phi(x, g(v))$, is a saddle function if for each $j = 1, \ldots, m$ one of the following holds:

   - $g_j(v)$ is convex and $\phi$ is nonincreasing in $y_j$ for all $x \in \mathcal{X}$ and all $y \in \mathcal{Y}$.
   - $g_j(v)$ is concave and $\phi$ is nondecreasing in $y_j$ for all $x \in \mathcal{X}$ and all $y \in \mathcal{Y}$.

### 4.2 Conic representable saddle functions

Nemirovski and Juditsky propose a class of *conic representable* saddle functions which facilitate the automated dualization of saddle problems (Juditsky & Nemirovski, 2022). We will first introduce some terminology and notation, and then describe the class of conic representable saddle functions.

**Notation.** We use the notation $\phi(x, y) : \mathcal{X} \times \mathcal{Y} \subseteq \mathbf{R}^{n \times m} \to \mathbf{R}$ to denote a saddle function which is convex in $x$ and concave in $y$. Let $K_x$, $K_y$ and $K$ be members of a collection $\mathcal{K}$ of closed, convex, and pointed cones with nonempty interiors in Euclidean spaces such that $\mathcal{K}$ contains a nonnegative ray, is closed with respect to taking finite direct products of its members, and is closed with respect to passing from a cone to its dual. We denote conic membership $z \in K$ by $z \succeq_K 0$. We call a set $\mathcal{X} \subseteq \mathbf{R}^n$ $\mathcal{K}$-representable if there exist constant matrices $A$ and $B$, a constant vector $c$, and a cone $K \in \mathcal{K}$ such that

$$\mathcal{X} = \{x \mid \exists u : Ax + Bu \preceq_K c\}.$$

CVXPY (Diamond & Boyd, 2016) can implement a function $f$ exactly when its epigraph $\{(x, u) \mid f(x) \leq u\}$ is $\mathcal{K}$-representable.

**Conic representable saddle functions.** Let $\mathcal{X}$ and $\mathcal{Y}$ be nonempty and possessing $\mathcal{K}$-representations

$$\mathcal{X} = \{x \mid \exists u : Ax + Bu \preceq_K c\}, \quad \mathcal{Y} = \{y \mid \exists v : Cy + Dv \preceq_K e\}.$$

A saddle function $\phi(x, y) : \mathcal{X} \times \mathcal{Y} \to \mathbf{R}$ is $\mathcal{K}$-representable if there exist constant matrices $P$, $Q$, $R$, constant vectors $p$ and $s$ and a cone $K \in \mathcal{K}$ such that for each $x \in \mathcal{X}$ and $y \in \mathcal{Y}$,

$$\phi(x, y) = \inf_{f, t, u} \{f^T y + t \mid Pf + tp + Qu + Rx \preceq_K s\}.$$

Here $f$ is a vector of the same dimension as $y$, $t$ is a scalar, and $u$ is a vector. This definition generalizes simple class of bilinear saddle functions. See Juditsky & Nemirovski (2022) for much more detail.

**Automated dualization.** Suppose we have a $\mathcal{K}$-representable saddle function $\phi$ as above. The conic form allows us to derive a dualized representation of the saddle extremum function

$$\Phi(x) = \sup_{y \in \mathcal{Y}} \phi(x, y)$$

which again admits a tractable conic form, meaning that it can be represented in a DSL like CVXPY. Specifically,

$$
\begin{aligned}
\Phi(x) &= \sup_{y \in \mathcal{Y}} \phi(x, y) \\
&= \sup_{y \in \mathcal{Y}} \inf_{f, t, u} \left\{ f^T y + t \mid Pf + tp + Qu + Rx \preceq_K s \right\} \\
&= \inf_{f, t, u} \left\{ \sup_{y \in \mathcal{Y}} \left( f^T y + t \right) \,\middle|\, Pf + tp + Qu + Rx \preceq_K s \right\} \quad (12) \\
&= \inf_{f, t, u} \left\{ \sup_{y \in \mathcal{Y}} \left( f^T y \right) + t \,\middle|\, Pf + tp + Qu + Rx \preceq_K s \right\} \\
&= \inf_{f, t, u} \left\{ \inf_{\lambda} \left\{ \lambda^T e \,\middle|\, \begin{matrix} C^T \lambda = f, \ D^T \lambda = 0 \\ \lambda \succeq_{K^*} 0 \end{matrix} \right\} + t \,\middle|\, Pf + tp + Qu + Rx \preceq_K s \right\} \quad (13)
\end{aligned}
$$

where in equation 12 we use Sion's minimax theorem (Sion, 1958) to reverse the inf and sup, and in equation 13 we invoke strong duality to replace the supremum over $y$ with an infimum over $\lambda$. Concretely, strong duality and the conic structure allow us to equate

$$\sup_{y} \left\{ f^T y \mid Cy + Dv \preceq_K e \right\} = \inf_{\lambda} \left\{ \lambda^T e \mid C^T \lambda = f, \ D^T \lambda = 0, \ \lambda \succeq_{K^*} 0 \right\},$$

where $K^*$ is the dual cone of $K$. This is exactly the automated dualization made possible by the conic representable form of $\phi$ (which DSP provides). Given the conic representation of $\phi$, the dualized form is obtained via the explicit formula given in equation 13.

The final line implies a conic representation of the epigraph of $\Phi(x)$,

$$\{(x, u) \mid \Phi(x) \le u\} = \left\{ (x, u) \ \middle| \ \exists \lambda, f, t, u : \begin{array}{l} \lambda^T e + t \le u \\ C^T \lambda = f, \ D^T \lambda = 0, \ \lambda \succeq_{K^*} 0 \\ Pf + tp + Qu + Rx \preceq_K s \end{array} \right\},$$

which is tractable and can be implemented in a DSL like CVXPY. This transformation is exact, and so there is no notion of approximation error or optimality gap arising from the dualization procedure.

**A mathematical nuance.** Switching the inf and sup in equation 12 requires Sion's theorem to hold. A sufficient condition for Sion's theorem to hold is that the set $\mathcal{Y}$ is compact. However, the min and max can be exchanged even if $\mathcal{Y}$ is not compact. Then, due to the max-min inequality

$$\max_{y \in \mathcal{Y}} \min_{x \in \mathcal{X}} f(x, y) \le \min_{x \in \mathcal{X}} \max_{y \in \mathcal{Y}} f(x, y),$$

the equality in equation 13 is replaced with a less than or equal to, and we obtain a convex restriction. Thus, if a user creates a problem involving an SE function (as opposed to a saddle point problem only containing saddle functions in the objective), then DSP guarantees that the problem generated is a restriction. This means that the variables returned are feasible and the returned optimal value is an upper bound on the optimal value for the user's problem.

**Obtaining convex and concave saddle point coordinates.** One challenge that arises in transforming the mathematical concept of conic representable saddle functions into a practical implementation is that the automated dualization removes the concave variable from the problem. Additionally, the procedure relies on the technical conditions such as compactness, which we believe should not be exposed in a user interface. We now address these points.

In our implementation, a saddle problem with an SE function in the objective is solved by applying the above automatic dualization to both the objective $\phi$ and $-\phi$ and then solving each resulting convex problem. Note that $\phi(x, y)$ is convex in $x$ and concave in $y$, while $-\phi(x, y)$ is concave in $x$ and convex in $y$. We do so in order to obtain both the convex and concave components of the saddle point, since the dualization removes the concave variable. To see this, note that equation 13 contains $x$ but not $y$ (and the opposite holds for the negated problem). The saddle problem is only reported as solved if the optimal value of the problem with objective $\phi$, $u$, is within a numerical tolerance of the negated optimal value of the problem with objective $-\phi$, $-l$. If this holds, this actually implies that

$$\max_{y \in \mathcal{Y}} \min_{x \in \mathcal{X}} \phi(x, y) = \min_{x \in \mathcal{X}} \max_{y \in \mathcal{Y}} \phi(x, y),$$

*i.e.*, equation 12 was valid, even if for example $\mathcal{Y}$ is not compact. To see this, note that solving for $\phi$ as well as $-\phi$ results in an upper and a lower bound on the optimal value of the saddle point problem,

$$\max_{y \in \mathcal{Y}} \min_{x \in \mathcal{X}} \phi(x, y) \le \min_{x \in \mathcal{X}} \max_{y \in \mathcal{Y}} \phi(x, y) \le u, \quad \text{and} \quad \max_{x \in \mathcal{X}} \min_{y \in \mathcal{Y}} -\phi(x, y) \le \min_{y \in \mathcal{Y}} \max_{x \in \mathcal{X}} -\phi(x, y) \le -l.$$

Using symmetry and combining the above inequalities, we obtain

$$l \le \max_{y \in \mathcal{Y}} \min_{x \in \mathcal{X}} \phi(x, y) \le \min_{x \in \mathcal{X}} \max_{y \in \mathcal{Y}} \phi(x, y) \le u.$$

Suppose now that $l = u$. Note that since

$$\phi(x^\star, y^\star) \le \max_{y \in \mathcal{Y}} \phi(x^\star, y) = u, \quad \text{and} \quad \phi(x^\star, y^\star) \ge \min_{x \in \mathcal{X}} \phi(x, y^\star) = l,$$

we have that $\phi(x^\star, y^\star) = l = u$. That is, the pair $(x^\star, y^\star)$ obtains the optimal value of the saddle point problem. All that remains is to verify that this pair satisfies the saddle point property. We have that $\phi(x^\star, y) \leq \phi(x^\star, y^\star)$ for all $y \in \mathcal{Y}$, since otherwise $u = \phi(x^\star, y^\star) < \max_{y \in \mathcal{Y}} \phi(x^\star, y) = u$, a contradiction. Similarly, $\phi(x, y^\star) \geq \phi(x^\star, y^\star)$ for all $x \in \mathcal{X}$. Taken together, these inequalities state that $(x^\star, y^\star)$ is a saddle point, since

$$\phi(x^\star, y) \leq \phi(x^\star, y^\star) \leq \phi(x, y^\star), \quad \forall x \in \mathcal{X}, y \in \mathcal{Y}.$$

Thus, a user need not concern themselves with the compactness of $\mathcal{Y}$ (or any other sufficient condition for Sion's theorem) when using DSP to find a saddle point; if a saddle point problem is solved, then the saddle point property is guaranteed to hold. This mathematical insight extends the work of Juditsky & Nemirovski (2022), which assumes compactness of $\mathcal{Y}$, allowing users who might be unfamiliar with this technical restriction to use DSP.

## 5 Implementation

In this section we describe our Python implementation of the concepts and methods described in §4, which we also call DSP. It can be accessed online under an open source license at `https://github.com/cvxgrp/dsp`. DSP works with CVXPY (Diamond & Boyd, 2016), an implementation of a DSL for convex optimization based on DCP. We use the term DSP in two different ways. We use it to refer to the mathematical concept of disciplined saddle programming, and also our specific implementation; which is meant should be clear from the context. The term DSP-compliant refers to a function or expression that is constructed according to the DSP composition rules given in §5.2. It can also refer to a problem that is constructed according to these rules. In the code snippets below, we use the prefix `cp` to indicate functions and classes from CVXPY. (We give functions and classes from DSP without prefix, whereas they would likely have a prefix such as `dsp` in real code.)

### 5.1 Atoms

Saddle functions in DSP are created from fundamental building blocks or atoms. These building blocks extend the atoms from CVXPY Diamond & Boyd (2016). In CVXPY, atoms are either jointly convex or concave in all their variables, but in DSP, atoms are (jointly) convex in a subset of the variables and (jointly) concave in the remaining variables. We describe some DSP atoms below. The listing is not exhaustive, and additional atoms can be added as necessary.

**Inner product.** The atom `inner(x,y)` represents the inner product $x^T y$. Since either $x$ or $y$ could represent the convex variable, we adopt the convention in DSP that the first argument of `inner` is the convex variable. According to the DSP rules, both arguments to `inner` must be affine, and the variables they depend on must be disjoint.

**Saddle inner product.** The atom `saddle_inner(F, G)` corresponds to the function $F(x)^T G(y)$, where $F$ and $G$ are vectors of nonnegative and respectively elementwise convex and concave functions. It is DSP-compliant if $F$ is DCP convex and nonnegative and $G$ is DCP concave. If the function $G$ is not DCP nonnegative, then the DCP constraint `G >= 0` is attached to the expression. This is analogous to how the DCP constraint `x >= 0` is added to the expression `cp.log(x)`. As an example consider

$$f = \texttt{saddle\_inner(cp.square(x), cp.log(y))}.$$

This represents the saddle function

$$f(x, y) = x^2 \log y - I(y \geq 1),$$

where $I$ is the $\{0, \infty\}$ indicator function of its argument.

**Weighted $\ell_2$ norm.** The `weighted_norm2(x, y)` atom represents the saddle function $\left(\sum_{i=1}^n y_i x_i^2\right)^{1/2}$, with $y \geq 0$. It is DSP-compliant if `x` is either DCP affine or both convex and nonnegative, and $y$ is DCP concave. Here too, the constraint `y >= 0` is added if $y$ is not DCP nonnegative.

**Weighted log-sum-exp.** The `weighted_log_sum_exp(x, y)` atom represents the saddle function $\log\left(\sum_{i=1}^{n} y_i \exp x_i\right)$, with $y \geq 0$. It is DSP-compliant if `x` is DCP convex, and $y$ is DCP concave. The constraint `y >= 0` is added if $y$ is not DCP nonnegative.

**Quasi-semidefinite quadratic form.** The `quasidef_quad_form(x, y, P, Q, S)` atom represents the function

$$f(x,y) = \left[\begin{array}{c} x \\ y \end{array}\right]^T \left[\begin{array}{cc} P & S \\ S^T & Q \end{array}\right] \left[\begin{array}{c} x \\ y \end{array}\right],$$

where the matrix is quasi-semidefinite, *i.e.*, $P \in \mathbf{S}_+^n$ and $-Q \in \mathbf{S}_+^n$. It is DSP-compliant if $x$ is DCP affine and $y$ is DCP affine.

**Quadratic form.** The `saddle_quad_form(x, Y)` atom represents the function $x^T Y x$, where $Y$ is a PSD matrix. It is DSP-compliant if $x$ is DCP affine, and $Y$ is DCP PSD.

## 5.2 Calculus rules

The atoms can be combined according to the calculus described below to form expressions that are DSP-compliant. For example, saddle functions can be added or scaled. DCP-compliant convex and concave expressions are promoted to saddle functions with no concave or convex variables, respectively. For example, with variables `x`, `y`, and `z`, the expression

```
f = 2.5 * saddle_inner(cp.square(x), cp.log(y)) + cp.minimum(y,1) - z
```

is DSP-compliant, with convex variable `x`, concave variable `y`, and affine variable `z`.

Calling the `is_dsp` method of an expression returns `True` if the expression is DSP-compliant. The methods `convex_variables`, `concave_variables`, and `affine_variables`, list the convex, concave, and affine variables, respectively. The convex variables are those that could only be convex, and similarly for concave variables. We refer to the convex variables as the unambiguously convex variables, and similarly for the concave variables. The three lists of variables gives a partition of all the variables the expression depends on. For the expression above, `f.is_dsp()` evaluates as `True`, `f.convex_variables()` returns the list `[x]`, `f.concave_variables()` returns the list `[y]`, and `f.affine_variables()` returns the list `[z]`. Note that the role of `z` is ambiguous in the expression, since it could be either a convex or concave variable.

**No mixing variables rule.** The DSP rules prohibit mixing of convex and concave variables. For example if we add two saddle expressions, no variable can appear in both its convex and concave variable lists.

**DSP-compliance is sufficient but not necessary to be a saddle function.** Recall that if an expression is DCP convex (concave), then it is convex (concave), but the converse is false. For example, the expression `cp.sqrt(1 + cp.square(x))` represents the convex function $\sqrt{1 + x^2}$, but is not DCP. But we can express the same function as `cp.norm2(cp.hstack([1, x]))`, which is DCP. The same holds for DSP and saddle function: If an expression is DSP-compliant, then it represents a saddle function; but it can represent a saddle function and not be DSP-compliant. As with DCP, such an expression would need to be rewritten in DSP-compliant form, to use any of the other features of DSP (such as a solution method). As an example, the expression `x.T @ C @ y` represents the saddle function $x^T C y$, but is not DSP-compliant. The same function can be expressed as `inner(x, C @ y)`, which is DSP-compliant. While this restrictive syntax is an inherent limitation of disciplined convex programming in general, it is required for any parser based on the DSP composition rules.

When there are affine variables in a DSP-compliant expression, it means that those variables could be considered either convex or concave; either way, the function is a saddle function.

**Example.** The code below defines the bi-linear saddle function $f(x,y) = x^T C y$, the objective of a matrix game, with $x$ the convex variable and $y$ the concave variable.

Creating a saddle function.

```python
from dsp import *  # notational convenience
import cvxpy as cp
import numpy as np

x = cp.Variable(2)
y = cp.Variable(2)
C = np.array([[1, 2], [3, 1]])

f = inner(x, C @ y)

f.is_dsp()  # True

f.convex_variables()  # [x]
f.concave_variables()  # [y]
f.affine_variables()  # []
```

Lines 1–3 import the necessary packages (which we will use but not show in the sequel). In lines 5–7, we create two CVXPY variables and a constant matrix. In line 9 we construct the saddle function `f` using the DSP atom `inner`. Both its arguments are affine, so this matches the DSP rules. In line 11 we check if `saddle_function` is DSP-compliant, which it is. In lines 13–15 we call functions that return lists of the convex, concave, and affine variables, respectively. The results of lines 13–15 might seem odd, but recall that `inner` marks its first argument as convex and its second as concave.

### 5.3 Saddle point problems

**Saddle point problem objective.** To construct a saddle point problem, we first create an objective using

$$\texttt{obj = MinimizeMaximize(f)},$$

where `f` is a CVXPY expression. The objective `obj` is DSP-compliant if the expression `f` is DSP-compliant. This is analogous to the CVXPY contructors `cp.Minimize(f)` and `cp.Maximize(f)`, which create objectives from expressions.

**Saddle point problem.** A saddle point problem is constructed using

$$\texttt{prob = SaddlePointProblem(obj, constraints, cvx\_vars, ccv\_vars)}$$

Here, `obj` is a `MinimizeMaximize` objective, `constraints` is a list of constraints, `cvx_vars` is a list of convex variables and `ccv_vars` is a list of concave variables. The objective must be DSP-compliant for the problem to be DSP-compliant. We now describe the remaining conditions under which the constructed problem is DSP-compliant.

Each constraint in the list must be DCP, and can only involve convex variables or concave variables; convex and concave variables cannot both appear in any one constraint. The list of convex and concave variables partitions all the variables that appear in the objective or the constraints. In cases where the role of a variable is unambiguous, it is inferred, and does not need to be in either list. For example with the objective

$$\texttt{MinimizeMaximize(weighted\_log\_sum\_exp(x, y) + cp.exp(u) + cp.log(v) + z)},$$

`x` and `u` must be convex variables, and `y` and `v` must be concave variables, and so do not need to appear in the lists used to construct a saddle point problem. The variable `z`, however, could be either a convex or concave variable, and so must appear in one of the lists.

The role of a variable can also be inferred from the constraints: Any variable that appears in a constraint with convex (concave) variables must also be convex (concave). With the objective above, the constraint

`z + v <= 1` would serve to classify `z` as a concave variable. With this constraint, we could pass empty variable lists to the saddle point constructor, since the roles of all variables can be inferred. When the roles of all variables are unambiguous, the lists are optional.

The roles of the variables in a saddle point problem `prob` can be found by calling `prob.convex_variables()` and `prob.concave_variables()`, which return lists of variables, and is a partition of all the variables appearing in the objective or constraints. This is useful for debugging, to be sure that DSP agrees with you about the roles of all variables. A DSP-compliant saddle point problem must have an empty list of affine variables. (If it did not, the problem would be ambiguous.)

**Solving a saddle point problem.** The `solve()` method of a `SaddlePointProblem` object canonicalizes and solves the problem. This involves checking the objective and constraints for DSP-compliance. The conic representation of the problem is obtained, which involves setting up an auxiliary problem and compiling it using CVXPY. Then, the dualization is carried out, which results in another CVXPY problem which is then solved to yield the objective value. This has the side effect of setting all convex variables' `.value` attribute. To also obtain the values of the concave variables, the saddle point problem is solved again with a negated objective and the roles of the minimization and maximization variables reversed. We emphasize that as DSP acts as a compiler, it does not implement any optimization algorithms itself, but rather relies on the solvers accessible through CVXPY.

**Example.** Here we create and solve a matrix game, continuing the example above where `f` was defined. We do not need to pass in lists of variables since their roles can be inferred.

Creating and solving a matrix game.

```python
obj = MinimizeMaximize(f)
constraints = [x >= 0, cp.sum(x) == 1, y >= 0, cp.sum(y) == 1]
prob = SaddlePointProblem(obj, constraints)

prob.is_dsp()  # True
prob.convex_variables()  # [x]
prob.concave_variables()  # [y]
prob.affine_variables()  # []

prob.solve()  # solves the problem
prob.value  # 1.6666666666666667
x.value  # array([0.66666667, 0.33333333])
y.value  # array([0.33333333, 0.66666667])
```

### 5.4 Saddle extremum functions

**Local variables.** An SE function has one of the forms

$$G(x) = \sup_{y \in \mathcal{Y}} f(x, y) \quad \text{or} \quad H(y) = \inf_{x \in \mathcal{X}} f(x, y),$$

where $f$ is a saddle function. Note that $y$ in the definition of $G$, and $x$ in the definition of $H$, are local or dummy variables, understood to have no connection to any other variable. Their scope extends only to the definition, and not beyond.

To express this subtlety in DSP, we use the class `LocalVariable` to represent these dummy variables. The variables that are maximized over (in a saddle max function) or minimized over (in a saddle min function) must be declared using the `LocalVariable()` constructor. Any `LocalVariable` in an SE function cannot appear in any other SE function.

**Constructing SE functions.** We construct SE functions in DSP using

$$\texttt{saddle\_max(f, constraints)} \quad \text{or} \quad \texttt{saddle\_min(f, constraints)}.$$

Here, `f` is a CVXPY scalar expression, and `constraints` is a list of constraints. We now describe the rules for constructing a DSP-compliant SE function.

If a `saddle_max` is being constructed, `f` must be DSP-compliant, and the function's concave variables, and all variables appearing in the list of constraints, must be `LocalVariable`s, while the function's convex variables must all be regular `Variable`s. A similar rule applies for `saddle_min`.

The list of constraints is used to specify the set over which the sup or inf is taken. Each constraint must be DCP-compliant, and can only contain `LocalVariable`s.

With `x` a `Variable`, `y_loc` a `LocalVariable`, `z_loc` a `LocalVariable`, and `z` a `Variable`, consider the following two SE functions:

```
1 f_1 = saddle_max(inner(x, y_loc) + z, [y_loc <= 1])
2 f_2 = saddle_max(inner(x, y_loc) + z_loc, [y_loc <= 1, z_loc <= 1])
```

Both are DSP-compliant. For the first, calling `f_1.convex_variables()` would return `[x, z]`, and calling `f_1.concave_variables()` would return `[y_loc]`. For the second, calling `f_2.convex_variables()` would return `[x]`, and `f_2.concave_variables()` return `[y_loc, z_loc]`.

Let `y` be a `Variable`. Both of the following are not DSP-compliant:

```
1 f_3 = saddle_max(inner(x, y_loc) + z, [y_loc <= 1, z <= 1])
2 f_4 = saddle_max(inner(x, y) + z_loc, [y_loc <= 1, z_loc <= 1])
```

The first is not DSP-compliant because `z` is not a `LocalVariable`, but appears in the constraints. The second is not DSP-compliant because `y` is not a `LocalVariable`, but appears as a concave variable in the saddle function.

**SE functions are DCP.** When they are DSP-compliant, a `saddle_max` is a convex function, and a `saddle_min` is a concave function. They can be used anywhere in CVXPY that a convex or concave function is appropriate. You can add them, compose them (in appropriate ways), use them in the objective or either side of constraints (in appropriate ways).

**Examples.** Now we provide full examples demonstrating construction of a `saddle_max`, which we can use to solve the matrix game described in §5.3 as a saddle problem involving an SE function.

Creating a saddle max.

```
1  # Creating variables
2  x = cp.Variable(2)
3
4  # Creating local variables
5  y_loc = LocalVariable(2)
6
7  # Convex in x, concave in y_loc
8  f = saddle_inner(C @ x, y_loc)
9
10 # maximizes over y_loc
11 G = saddle_max(f, [y_loc >= 0, cp.sum(y_loc) == 1])
```

Note that `G` is a CVXPY expression. Constructing a `saddle_min` works exactly the same way.

### 5.5 Saddle problems

A saddle problem is a convex problem that uses SE functions. To be DSP-compliant, the problem must be DCP (which implies all SE functions are DSP-compliant). When you call the solve method on a saddle problem involving SE functions, and the solve is successful, then all variables' `.value` fields are overwritten with optimal values. This includes `LocalVariable`s that the SE functions maximized or minimized over; they are assigned to the value of *a particular* maximizer or minimizer of the SE function at the value of the non-local variables, with no further guarantees.

**Example.**    We continue our example from §5.4 and solve the matrix game using either a saddle max.

Creating and solving a saddle problem using a saddle max to solve the matrix game.

```
1  prob = cp.Problem(cp.Minimize(G), [x >= 0, cp.sum(x) == 1])
2
3  prob.is_dsp()  # True
4
5  prob.solve()  # solving the problem
6  prob.value  # 1.6666666666666667
7  x.value  # array([0.66666667, 0.33333333])
```

## 6    Examples

In this section we give numerical examples, taken from §3, showing how to create DSP-compliant problems. The specific problem instances we take are small, since our main point is to show how easily the problems can be specified in DSP. But DSP will scale to far larger problem instances. Again, code and data for these examples are available at `https://github.com/cvxgrp/dsp`.

### 6.1    Robust bond portfolio construction

Our first example is the robust bond portfolio construction problem described in §3.1. We consider portfolios of $n = 20$ bonds, over a period $T = 60$ half-years, *i.e.*, 30 years. The bonds are taken as representative ones in a global investment grade bond portfolio; for more detail, see Luxenberg et al. (2022). The payments from the bonds are given by $C \in \mathbf{R}^{20 \times 60}$, with cash flow of bond $i$ in period $t$ denoted $c_{i,t}$. The goal is to choose holdings $h \in \mathbf{R}^{20}_+$, with the portfolio constraint set $\mathcal{H}$ given by

$$\mathcal{H} = \{h \mid h \geq 0, \ p^T h = B\},$$

*i.e.*, the investments must be nonnegative and have a total value (budget) $B$, which we take to be \$100. Here $p \in \mathbf{R}^{20}_+$ denotes the price of the bonds on September 12, 2022. The portfolio objective is

$$\phi(h) = \frac{1}{2}\|(h - h^{\mathrm{mkt}}) \circ p\|_1,$$

where $h^{\mathrm{mkt}} \in \mathbf{R}^{20}_+$ is the market portfolio scaled to a value of \$100, and $\circ$ denotes Hadamard or elementwise multiplication. This is called the turn-over distance, since it tells us how much we would need to buy and sell to convert our portfolio to the market portfolio.

The yield curve set $\mathcal{Y}$ is described in terms of perturbations to the nominal or current yield curve $y^{\mathrm{nom}} \in \mathbf{R}^{60}$, which is the yield curve on September 12, 2022. We take

$$\mathcal{Y} = \left\{ y^{\mathrm{nom}} + \delta \ \middle| \ \|\delta\|_\infty \leq \delta^{\mathrm{max}}, \ \|\delta\|_1 \leq \kappa, \ \sum_{t=1}^{T-1}(\delta_{t+1} - \delta_t)^2 \leq \omega \right\}.$$

We interpret $\delta \in \mathbf{R}^{60}$ as a shock to the yield curve, which we limit elementwise, in absolute sum, and in smoothness. The specific parameter values are given by

$$\delta^{\mathrm{max}} = 0.02, \quad \kappa = 0.9, \quad \omega = 10^{-6}.$$

In the robust bond portfolio problem equation 10 we take $V^{\text{lim}} = 90$, that is, the worst case value of the portfolio cannot drop below \$90 for any $y \in \mathcal{Y}$.

We solve the problem using the following code, where we assume the cash flow matrix `C`, the price vector `p`, the nominal yield curve `y_nom`, and the market portfolio `h_mkt` are defined.

Robust bond portfolio construction.

```
1  # Constants and parameters
2  n, T = C.shape
3  delta_max, kappa, omega = 0.02, 0.9, 1e-6
4  B = 100
5  V_lim = 90
6
7  # Creating variables
8  h = cp.Variable(n, nonneg=True)
9
10 delta = LocalVariable(T)
11 y = y_nom + delta
12
13 # Objective
14 phi = 0.5 * cp.norm1(cp.multiply(h, p) - cp.multiply(h_mkt, p))
15
16 # Creating saddle min function
17 V = 0
18 for i in range(n):
19     t_plus_1 = np.arange(T) + 1  # Account for zero-indexing
20     V += saddle_inner(cp.exp(cp.multiply(-t_plus_1, y)), h[i] * C[i])
21
22 Y = [
23     cp.norm_inf(delta) <= delta_max,
24     cp.norm1(delta) <= kappa,
25     cp.sum_squares(delta[1:] - delta[:-1]) <= omega,
26 ]
27
28 V_wc = saddle_min(V, Y)
29
30 # Creating and solving the problem
31 problem = cp.Problem(cp.Minimize(phi), [h @ p == B, V_wc >= V_lim])
32 problem.solve()  # 15.32
```

We first define the constants and parameters in lines 2–5, before creating the variable for the holdings `h` in line 8, and the `LocalVariable delta`, which gives the yield curve perturbation, in line 10. In line 11 we define `y` as the sum of the current yield curve `y_nom` and the perturbation `delta`. The objective function is defined in line 14. Lines 17–20 define the saddle function `V` via the `saddle_inner` atom. The yield uncertainty set `Y` is defined in lines 22–26, and the worst case portfolio value is defined in line 25 using `saddle_min`. We use the concave expression `saddle_min` to create and solve a CVXPY problem in lines 31–32.

Table 1 summarizes the results. The nominal portfolio is the market portfolio, which has zero turn-over distance to the market portfolio, *i.e.*, zero objective value. This nominal portfolio, however, does not satisfy the worst-case portfolio value constraint, since there are yield curves in $\mathcal{Y}$ that cause the portfolio value to drop to around \$87, less than our limit of \$90. The solution of the robust problem has turn-over distance \$15.32, and satisfies the constraint that the worst-case value be at least \$90.

|  | Nominal portfolio | Robust portfolio |
|---|---|---|
| Turn-over distance | $0.00 | $15.32 |
| Worst-case value | $86.99 | $90.00 |

Table 1: Turn-over distance and worst-case value for the nominal (market) portfolio and the robust portfolio. The nominal portfolio does not meet our requirement that the worst-case value be at least $90.

### 6.2 Model fitting robust to data weights

We consider an instance of the model fitting problem described in §3.2. We use the well known Titanic data set (Harrell Jr. & Cason, 2017), which gives several attributes for each passenger on the ill-fated Titanic voyage, including whether they survived. A classifier is fit to predict survival based on the features sex, age (binned into three groups, 0–26, 26–53, and 53–80), and class (1, 2, or 3). These features are encoded as a Boolean vector $a_i \in \mathbf{R}^7$. The label $y_i = 1$ means passenger $i$ survived, and $y_i = -1$ otherwise. There are 1046 examples, but we fit our model using only the $m = 50$ passengers who embarked from Queenstown, one of three ports of embarkation. This is a somewhat non-representative sample; for example, the survival rate among Queenstown departures is 26%, whereas the overall survival rate is 40.8%. This is a common situation in machine learning, where the distribution of labels in the training data does not match that of the test dataset (known as label shift), for which we seek a robust solution.

We seek a linear classifier $\hat{y}_i = \text{sign}(a_i^T \theta + \beta_0)$, where $\theta \in \mathbf{R}^7$ is the classifier parameter vector and $\beta_0 \in \mathbf{R}$ is the bias. The hinge loss and $\ell_2$ regularization are used, given by

$$\ell_i(\theta) = \max(0, 1 - y_i a_i^T \theta), \qquad r(\theta) = \eta \|\theta\|_2^2,$$

with $\eta = 0.05$.

The data is weighted to partially correct for the different survival rates for our training set (26%) and the whole data set (40.8%). To do this we set $w_i = z_1$ when $y_i = 1$ and $w_i = z_2$ when $y_i = -1$. We require $w \geq 0$ and $\mathbf{1}^T w = 1$, and

$$0.408 - 0.05 \leq \sum_{y_i=1} w_i \leq 0.408 + 0.05.$$

Thus $\mathcal{W}$ consists of weights on the Queenstown departure samples that correct the survival rate to within 5% of the overall survival rate.

The code shown below solves this problem, where we assume the data matrix is already defined as `A_train` (with rows $a_i^T$), the survival label vector is defined as `y_train`, and the indicator of survival in the training set is defined as `surv`.

Model fitting robust to data weights.

```python
# Constants and parameters
m, n = A_train.shape
inds_0 = surv == 0
inds_1 = surv == 1
eta = 0.05

# Creating variables
theta = cp.Variable(n)
beta_0 = cp.Variable()
weights = cp.Variable(m, nonneg=True)
surv_weight_0 = cp.Variable()
surv_weight_1 = cp.Variable()

# Defining the loss function and the weight constraints
y_hat = A_train @ theta + beta_0
```

|  | Nominal classifier | Robust classifier |
|---|---|---|
| Train accuracy | 82.0% | 80.0% |
| Test accuracy | 76.0% | 78.6% |

Table 2: Nominal and worst-case objective values for classification and robust classification models.

```
16  loss = cp.pos(1 - cp.multiply(y_train, y_hat))
17  objective = MinimizeMaximize(saddle_inner(loss, weights)
18      + eta * cp.sum_squares(theta))
19
20  constraints = [
21      cp.sum(weights) == 1,
22      0.408 - 0.05 <= weights @ surv,
23      weights @ surv <= 0.408 + 0.05,
24      weights[inds_0] == surv_weight_0,
25      weights[inds_1] == surv_weight_1,
26  ]
27
28  # Creating and solving the problem
29  problem = SaddlePointProblem(objective, constraints)
30  problem.solve()
```

After defining the constants and parameters in lines 2–5, we specify the variables for the model coefficient and the weights in lines 8–9 and 10–12, respectively. The loss function and regularizer which make up the objective are defined next in lines 15–18. The weight constraints are defined in lines 20–26. The saddle point problem is created and solved in lines 29 and 30.

The results are shown in table 2. We report the test accuracy on all samples in the dataset with a different port of embarkation than Queenstown (996 samples). We see that while the robust classification model has slightly lower training accuracy than the nominal model, it achieves a higher test accuracy, generalizing from the non-representative training data better than the nominal classifier, which uses uniform weights.

### 6.3 Robust Markowitz portfolio construction

We consider the robust Markowitz portfolio construction problem described in §3.4. We take $n = 6$ assets, which are the (five) Fama-French factors (Fama & French, 2015) plus a risk-free asset. The data is obtained from the Kenneth R. French data library (French, 2022), with monthly return data available from July 1963 to October 2022. The nominal return and risk are the empirical mean and covariance of the returns. (These obviously involve look-ahead, but the point of the example is how to specify and solve the problem with DSP, not the construction of a real portfolio.) We take parameters $\rho = 0.02$, $\eta = 0.2$, and risk aversion parameter $\gamma = 1$.

In the code, we use `mu` and `Sigma` for the mean and covariance estimates, respectively, and the parameters are denoted `rho`, `eta`, and `gamma`.

Robust Markowitz portfolio construction.

```
1  # Constants and parameters
2  n = len(mu)
3  rho, eta, gamma = 0.2, 0.2, 1
4
5  # Creating variables
6  w = cp.Variable(n, nonneg=True)
7
8  delta_loc = LocalVariable(n)
```

|                   | Nominal portfolio | Robust portfolio |
|-------------------|-------------------|------------------|
| Nominal objective | .295              | .291             |
| Robust objective  | .065              | .076             |

Table 3: Nominal and worst-case objective for the nominal and robust portfolios.

```python
 9  Sigma_perturbed = LocalVariable((n, n), PSD=True)
10  Delta_loc = LocalVariable((n, n))
11
12  # Creating saddle min function
13  f = w @ mu + saddle_inner(delta_loc, w) \
14      - gamma * saddle_quad_form(w, Sigma_perturbed)
15
16  Sigma_diag = Sigma.diagonal()
17  local_constraints = [
18      cp.abs(delta_loc) <= rho, Sigma_perturbed == Sigma + Delta_loc,
19      cp.abs(Delta_loc) <= eta * np.sqrt(np.outer(Sigma_diag, Sigma_diag))
20  ]
21
22  G = saddle_min(f, local_constraints)
23
24  # Creating and solving the problem
25  problem = cp.Problem(cp.Maximize(G), [cp.sum(w) == 1])
26  problem.solve()  # 0.076
```

We first define the constants and parameters, before creating the weights variable in line 6, and the local variables for the perturbations in lines 8–10. The saddle function for the objective is defined in line 13, followed by the constraints on the perturbations. Both are combined into the concave saddle min function, which is maximized over the portfolio constraints in lines 25–26.

The results are shown in table 3. The robust portfolio yields a slightly lower risk adjusted return of 0.291 compared to the nominal optimal portfolio with 0.295. But the robust portfolio attains a higher worst-case risk adjusted return of 0.076, compared to the nominal optimal portfolio which attains 0.065.

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
