# OpenReview forum: "Disciplined Saddle Programming"
_TMLR — Accepted by TMLR_

### Review · Reviewer_toyQ · 2023-08-29

**Summary Of Contributions:**

The paper titled "Disciplined Saddle Programming" introduces the concept of disciplined saddle programming (DSP), a domain-specific language (DSL) for specifying and solving saddle problems, which are convex-concave optimization problems arising in various fields like game theory, machine learning, and finance. The paper's primary contribution is the development of a methodology that automates the dualization trick, which converts saddle problems into standard convex optimization problems. DSP allows users to easily formulate and solve saddle problems, even without expertise in convex duality. The authors extend the idea of disciplined convex programming (DCP) to saddle problems and provide an open-source implementation of DSP.

**Audience:**

Yes

**Broader Impact Concerns:**

no broader impact concerns

**Claims And Evidence:**

Yes

**Requested Changes:**

**Implementation Details**: Expand upon the implementation details of DSP. Discuss its limitations, challenges, and potential improvements. Provide examples of benchmarks or case studies showcasing the performance and efficiency of the DSP implementation in comparison to manual approaches.

**Theoretical Background**: Expand Section 4 Disciplined saddle point programming. Elaborate on the theoretical foundations of DSP, specifically how the dualization process using conic duality is automated. This could involve providing illustrative examples or derivations to aid the reader's understanding.

**Strengths And Weaknesses:**

## Strengths:

- **Automated Dualization**: The paper's central contribution of automating the dualization trick for saddle problems fills a gap in the existing literature. This can significantly simplify problem formulation and reduce the risk of errors, making it accessible to a broader audience.

- **Domain-Specific Language (DSL)**: Introducing a dedicated DSL for specifying saddle problems, similar to disciplined convex programming (DCP) for convex problems, enhances usability. This approach abstracts away the complex mathematical steps involved in the dualization process.

- **Applications in Diverse Fields**: By highlighting the applications of saddle problems in game theory, machine learning, and finance, the paper showcases the practical relevance of the proposed methodology. This adds value to researchers and practitioners in these areas.

- **Open-Source Implementation**: The provision of an open-source implementation of DSP enhances reproducibility and allows other researchers to use and build upon the proposed approach.

## Weaknesses:

- **Scope of Implementation**: The paper briefly mentions implementing DSP as a Python package and using it in conjunction with CVXPY for convex optimization. However, the paper does not discuss potential limitations, challenges, or performance considerations associated with the implementation.

- **Theoretical Foundations**: While the paper mentions the theoretical underpinnings of DSP and its extension from disciplined convex programming (DCP), it might be beneficial to provide more in-depth explanations of how the dualization process is automated using conic duality.

---

> ### Author Response · Authors · 2023-09-08
> **Author Response to Reviewer toyQ**
>
> We thank the reviewer for their comments. We respond to the requested changes inline below.
>
> > Implementation Details: Expand upon the implementation details of DSP. Discuss its limitations, challenges, and potential improvements. Provide examples of benchmarks or case studies showcasing the performance and efficiency of the DSP implementation in comparison to manual approaches.
>
> Please see our performance comparison above regarding the benchmarks.
>
> As an avenue of future improvement, we now state that adding additional atoms to the implementation may be necessary.
>
> > Theoretical Background: Expand Section 4 Disciplined saddle point programming. Elaborate on the theoretical foundations of DSP, specifically how the dualization process using conic duality is automated. This could involve providing illustrative examples or derivations to aid the reader's understanding.
>
> Thank you for bringing this up. We have rewritten this section to be more explicit about the automatic dualization.

---

> > ### Author Response · Authors · 2024-01-18
> > **Summary of changes**
> >
> > In addition to concerns raised by several reviewers addressed jointly in our [general comment](https://openreview.net/forum?id=KhMLfEIoUm&noteId=tbZLFJJkii), we respond point-to-point to individual comments below.
> >
> > **toyQ:** Theoretical Background: Expand Section 4 Disciplined saddle point programming. Elaborate on the theoretical foundations of DSP, specifically how the dualization process using conic duality is automated. This could involve providing illustrative examples or derivations to aid the reader's understanding.
> >
> > **Author response:** We improved section 4 to improve clarity. Beyond fixing notational errors pointed out by reviewer s9NH, we added
> >
> > > Concretely, strong duality and the conic structure allow us to equate
> > $$
> > \\sup_{y}\\left\\{f^Ty \\middle| Cy+Dv\\preceq_K e \\right\\} =
> > \\inf_{\\lambda}\\left\\{\\lambda^Te \\middle| C^T\\lambda=f,\\; D^T\\lambda=0,\\; \\lambda\\succeq_{K^*}0\\right\\},
> > $$
> > where $K^*$ is the dual cone of $K$. This is exactly the automated dualization made possible by the conic representable form of $\phi$.
> >
> > and
> >
> > > This is exactly the automated dualization made possible by the conic representable form of $\phi$ (which DSP provides). Given the conic representation of $\phi$, the dualized form is obtained via the explicit formula given in equation 13.
> >
> > to better describe the idea behind the automated dualization procedure.
> >
> > We also expanded the implementational description of this process.

---

### Review · Reviewer_MvoZ · 2023-08-30

**Summary Of Contributions:**

The paper presents a disciplined way for automatically dualizing the saddle programming problem. It provides a detailed background for the saddle programming with examples covering many related fields and applications. It derives the method from scratch and describes in full details how the method can be implemented in CVXPY. Three numerical experiments demonstrate how the method applies to different contexts.

**Audience:**

Yes

**Broader Impact Concerns:**

No broader impact concerns.

**Claims And Evidence:**

Yes

**Requested Changes:**

Besides the above two points, I think the following two aspects can be some viable options for improving the manuscripts:

- Optimality gap: From the technical sections to the numerical experiments of the paper, there is not much mentioned about the optimality gap of the output solution from the proposed method. Like for the numerical experiments, the solution is only compared against the nominal solution which can be a trivial solution for application contexts of Section 6.1 and 6.2. In my opinion, an advantage over the nominal solution doesn't tell much about the quality of the given solution. As I understand, it should be not difficult (correct me if I am wrong here) to derive an optimality gap which can better demonstrate the quality of the solution. There can also be other ways to visualize and better assess the quality of the solution. The current comparison against the nominal solution seems too simple.

- Computational time: I understand that the focus of the paper is to provide a systematic and automatic way to convexify saddle point problem, and the computation time for solving a problem depends on the used solver and the intrinsic difficulty of the underlying problem. Yet some numerical experiments can be more helpful and insightful to ML audience. For example, for the reweighting ERM problem in Section 3.2, how different choices of the weight set affect the computational time, and how the computational time may scale with number of samples. These might be out of the scope of the current positioning of the paper, but can be of interests to people in related topics. Also, for financial applications, the computational time and the scalability is also of great importance in practice. Some demonstrations on the computational aspects of the numerical experiments in Section 6.1 and 6.3 can be helpful.

Minor comments:

I appreciate the clarity of the writing of the paper. There are still minor typos and I'd suggest the authors to proofread the paper again thoroughly:

- Page 3: $\lambda_n$ should be $lambda_m$.
- Page 3: the set $\mathcal{Y}$ should be $R^p\times R^{m}_+$.
- Page 4: As "a" more esoteric example
- Page 6: there is an extra comma after $sup\  y^\top x$
- Page 9: the norm of $\delta$ should be $||\cdot||$; and also what's the norm used here?
- Page 17: maybe specify the dimension of $\delta$ and $\delta=(\delta_1,...,\delta_T)$

At the end of this review, my sincere apologies to the authors and editor team for the delay in submitting my review :-)

**Strengths And Weaknesses:**

The paper is really well-written, intuitive, and easy to follow. Even for people not familiar with the saddle point problem, it gives enough background on how the saddle point problem is formulated from application contexts and how the proposed method can be programmed and implemented.

My main concern is on the following two points:

- (i) The interests of the work to general ML audience. In the examples and numerical experiments, i.e. Section 3 and Section 6, most of the examples are from finance or operations research applications, with the only exception of the weighted ERM problem in Section 3.2. This weighted ERM problem is no doubt an important topic for the distributional robust optimization or adversarial learning community in ML. But it seems this paper only demonstrates for the case where the weight choice set $\mathcal{W}$ is a polyhedron specified by linear constraints. In my opinion, it would make the paper more attractive if the numerical experiments consider more weight choice set such as a set specified by the $\chi$-square distance or the Wasserstein distance towards the uniform distribution, as this would better resonate with the “ML” applications.

- As a minor point, the numerical example in Section 6.2 is not quite for a robust ML purpose, but it is more of a simple label-shift adjustment to match the survival rate between the training set and the test set.

- (ii) The technical contribution of the work: From a technical viewpoint, it seems to me that the paper is more of the results Juditsky & Nemirovski (2022) and its contribution is more on the engineering side for a systematic way to implement the automatic dualization idea proposed therein. Please correct me if I miss anything; and I would suggest the authors to better clarify the technical contribution of the paper against that paper and the existing literature or whether the paper is positioned as one that purely contributes for an engineering implementation.

---

> ### Author Response · Authors · 2023-09-08
> **Author Response to Reviewer MvoZ**
>
> We thank the reviewer for their comments. We addressed the major points raised and respond to the requested changes inline below.
>
> > Optimality gap: From the technical sections to the numerical experiments of the paper, there is not much mentioned about the optimality gap of the output solution from the proposed method. Like for the numerical experiments, the solution is only compared against the nominal solution which can be a trivial solution for application contexts of Section 6.1 and 6.2. In my opinion, an advantage over the nominal solution doesn't tell much about the quality of the given solution. As I understand, it should be not difficult (correct me if I am wrong here) to derive an optimality gap which can better demonstrate the quality of the solution. There can also be other ways to visualize and better assess the quality of the solution. The current comparison against the nominal solution seems too simple.
>
> We did not expand on the optimality gap as the dualized problem yields the same
> objective value as the original problem. We have modified the paper to clarify this
> point. In all experiments, the solutions returned by the solvers are
> guaranteed to be optimal up to numerical precision. We hope this addresses your
> concern.
>
> > Computational time: I understand that the focus of the paper is to provide a systematic and automatic way to convexify saddle point problem, and the computation time for solving a problem depends on the used solver and the intrinsic difficulty of the underlying problem. Yet some numerical experiments can be more helpful and insightful to ML audience. For example, for the reweighting ERM problem in Section 3.2, how different choices of the weight set affect the computational time, and how the computational time may scale with number of samples. These might be out of the scope of the current positioning of the paper, but can be of interests to people in related topics. Also, for financial applications, the computational time and the scalability is also of great importance in practice. Some demonstrations on the computational aspects of the numerical experiments in Section 6.1 and 6.3 can be helpful.
>
> Please see our performance comparison above regarding the benchmarks.
>
> > Minor comments:
> I appreciate the clarity of the writing of the paper. There are still minor typos and I'd suggest the authors to proofread the paper again thoroughly: [...]
>
> Thank you for pointing out the typos. We have fixed them.

---

> > ### Author Response · Authors · 2024-01-18
> > **Summary of changes**
> >
> > In addition to concerns raised by several reviewers addressed jointly in our [general comment](https://openreview.net/forum?id=KhMLfEIoUm&noteId=tbZLFJJkii), we respond point-to-point to individual comments below.
> >
> > To make our work more relatable to the ML audience, we now include the example of constraining the Wassertein distance of the weights mentioned by the reviewer in our manuscript, which indeed is handled by our method. We now also mention the term label-shift adjustment to describe the numerical example in section 6.2 more clearly.
> >
> > **Mvoz:**
> > Optimality gap: From the technical sections to the numerical experiments of the paper, there is not much mentioned about the optimality gap of the output solution from the proposed method. Like for the numerical experiments, the solution is only compared against the nominal solution which can be a trivial solution for application contexts of Section 6.1 and 6.2. In my opinion, an advantage over the nominal solution doesn't tell much about the quality of the given solution. As I understand, it should be not difficult (correct me if I am wrong here) to derive an optimality gap which can better demonstrate the quality of the solution. There can also be other ways to visualize and better assess the quality of the solution. The current comparison against the nominal solution seems too simple.
> >
> > **Author response:** We clarified that the mathematical transformations are exact by adding the following sentence to our manuscript:
> > > This transformation is exact, and so there is no notion of approximation error or
> > optimality gap arising from the dualization procedure.
> >
> > Thus, the only notion of an optimality gap is in the resulting convex problem, which has the same interpretation as for any other convex problem.

---

### Review · Reviewer_7kLT · 2023-08-31

**Summary Of Contributions:**

This paper introduces and implements a domain-specific language called disciplined saddle programming (DSP). DSP is designed specifically for specifying and addressing convex-concave saddle problems. Building on the research by Juditsky & Nemirovski in 2022, DSP utilizes the concept of conic-representable saddle point programs and leverages the automatic dualization technique to reduce bilevel problems to single convex optimization using conic duality. The authors have implemented DSP as an open-source package, streamlining the process for users to formulate and solve saddle problems.

**Audience:**

Yes

**Broader Impact Concerns:**

No further concerns about the ethical implications.

**Claims And Evidence:**

Yes

**Requested Changes:**

1. The foundational ideas and methodologies for this work stem from Juditsky & Nemirovski's research in 2022 (https://doi.org/10.1080/10556788.2021.1928121). The primary contribution of this paper lies in its implementation of their methods into a user-friendly package for solving saddle point problems. However, the paper does not easily allow readers to assess the authors' engineering endeavors in translating the theory into a user-friendly toolkit. It would greatly enhance the reader's experience if the authors could provide more details or perhaps a dedicated section illustrating the engineering and developmental challenges.

2. Overall, the paper is easy to follow. However, I am mainly confused about the following point:
While the authors tried to derive a procedure to obtain both the convex and concave coordinates of the saddle point after reducing to a single convex problem, the representation of this process lacks clarity. A potential improvement for the authors would be to incorporate an implementation example for this procedure, as it plays a crucial role in enhancing the toolbox's practicality.

**Strengths And Weaknesses:**

1. Drawing from theoretical insights in existing literature, the authors have transformed the theory into a practical approach for specifying and solving convex-concave saddle problems. Additionally, the authors developed a user-friendly open-source package compatible with CVXPY. To my knowledge, this package is the first end-to-end toolbox designed for solving min-max problems.

2. The paper is also clearly written, and the authors have incorporated numerous illustrative examples from various domains, showcasing the applicability of DSP. As such, this paper can also serve as a comprehensive guide to the DSP package.

---

> ### Author Response · Authors · 2023-09-08
> **Author Response to Reviewer 7kLT**
>
> We thank the reviewer for their comments. We respond to the requested changes inline below.
>
> > The foundational ideas and methodologies for this work stem from Juditsky & Nemirovski's research in 2022 (https://doi.org/10.1080/10556788.2021.1928121). The primary contribution of this paper lies in its implementation of their methods into a user-friendly package for solving saddle point problems. However, the paper does not easily allow readers to assess the authors' engineering endeavors in translating the theory into a user-friendly toolkit. It would greatly enhance the reader's experience if the authors could provide more details or perhaps a dedicated section illustrating the engineering and developmental challenges.
>
> Thank you for this feedback. We have modified the paper to emphasize several
> implementation challenges, such as hiding compactness requirements from users,
> obtaining both $x$ and $y$ portions of saddle points, and local variables.
>
> > Overall, the paper is easy to follow. However, I am mainly confused about the following point: While the authors tried to derive a procedure to obtain both the convex and concave coordinates of the saddle point after reducing to a single convex problem, the representation of this process lacks clarity. A potential improvement for the authors would be to incorporate an implementation example for this procedure, as it plays a crucial role in enhancing the toolbox's practicality.
>
> Thank you for pointing this out. We have rewritten this section to be more
> explicit about why combining the two dualized problems optimal $x$ and $y$
> yields a saddle point when the optimal values match.

---

> > ### Author Response · Authors · 2024-01-18
> > **Summary of changes**
> >
> > In addition to concerns raised by several reviewers addressed jointly in our [general comment](https://openreview.net/forum?id=KhMLfEIoUm&noteId=tbZLFJJkii), we respond point-to-point to individual comments below.
> >
> > **7kLT:**
> > Overall, the paper is easy to follow. However, I am mainly confused about the following point: While the authors tried to derive a procedure to obtain both the convex and concave coordinates of the saddle point after reducing to a single convex problem, the representation of this process lacks clarity. A potential improvement for the authors would be to incorporate an implementation example for this procedure, as it plays a crucial role in enhancing the toolbox's practicality.
> >
> > **Author response:** We improved the description of the procedure, and it should now be easier to follow.
> >
> > > In our implementation, a saddle problem with an SE function in the objective is solved by applying the above automatic dualization to both the objective $\phi$ and $-\phi$ and then solving each resulting convex problem. Note that $\phi(x,y)$ is convex in $x$ and concave in $y$, while $-\phi(x,y)$ is concave in $x$ and convex in $y$. We do so in order to obtain both the convex and concave components of the saddle point, since the dualization removes the concave variable. To see this, note that equation 13 contains $x$ but not $y$ (and the opposite holds for the negated problem).
> >
> > We provide this detail to readers who are interested in the technical inner workings of DSP, noting that the practicality of the software is such that most user need not to concern themselves with these nuances. We also note that all of the numerical examples carry out this procedure behind the scenes: every time the solve method of a SaddlePointProblem object is called, the convex and concave variables value attributes are automatically set to optimal values without any action required by the user.

---

### Review · Reviewer_FqVv · 2023-09-01

**Summary Of Contributions:**

This paper introduces a domain-specific language (DSL) called disciplined saddle programming (DSP), catering to conic-representable saddle point programs. It is the first toolbox that can be directly used for these problems, where duality tricks can be used automatically. This paper contributes an open-source package that enables users to effectively tackle saddle problems without requiring any understanding of duality tricks.

**Audience:**

Yes

**Broader Impact Concerns:**

No broader impact concerns

**Claims And Evidence:**

Yes

**Requested Changes:**

It would be better to include a section to describe the challenge in the implementation and the comparison with other state-of-the-art methods.

**Strengths And Weaknesses:**

Strengths:
The paper is well-written and easy to follow.

**First open-source package:** This paper introduces a domain-specific language (DSL) called disciplined saddle programming (DSP). To the best of my knowledge, it is the first open source for solving saddle problems.

**User friendliness:**  The inclusion of numerous illustrative examples greatly enhances the reader's grasp of how to formulate and solve problems using DSP. Users can use it without knowledge of duality tricks and without the prior verification of the existence of saddle points.

Weakness: There are mainly two concerns:

**Implementation insight:** This paper provides a packaged implementation of Juditsky and Nemirovski’s work for conic representable saddle point programming. It would be better if you could elaborate more on the technical contribution to implementation/difficulties encountered in the implementation.

**Comparison with SOTA  methods:** As a solver, it would be beneficial to compare the time consumption and convergence performance with the current state-of-the-art methods. This empirical evidence would be valuable for assessing the real-world applicability and effectiveness of the proposed method.

---

> ### Author Response · Authors · 2023-09-08
> **Author Response to Reviewer FqVv**
>
> We thank the reviewer for their comments and respond to the requested changes inline below.
>
> > It would be better to include a section to describe the challenge in the implementation and the comparison with other state-of-the-art methods.
>
> For a performance comparison, please see our comment above.
> We describe in more detail the process involved when solving a saddle point
> problem and list additional atomic functions as an area of future improvement.

---

> > ### Author Response · Authors · 2024-01-18
> > **Summary of changes**
> >
> > Thank you again for your comments. We have added a [general comment](https://openreview.net/forum?id=KhMLfEIoUm&noteId=tbZLFJJkii) addressing our changes.

---

### Review · Reviewer_s9NH · 2023-09-01

**Summary Of Contributions:**

The main contributions are the following:
- The authors provide a very clear **introduction to saddle problems** and provide a **great intuition** on how to transform them into convex conic programs. It helps a lot to digest Juditsky and Nemiroski 2022 on which the theoretical part of the manuscript is based.
- Authors provide a **systematic way** of transforming a specific type of **constrained saddle** problem into a **convex program**, these types of problems are called "disciplined saddle programs"

**Audience:**

Yes

**Claims And Evidence:**

Yes

**Requested Changes:**

Most of my concerns are regarding the clarity of Part 4

Contributions:
- I would merge the 2 first points in the contribution (p3, section 1.2)

Clarity before part 4:
- I found the definition of the mathematical objects very hard to parse, I had to read them multiple times to correctly understand what they were precisely. In particular, I would recommend encapsulating the mathematical definitions of saddle functions, saddle points, saddle problems, saddle extremum function, and conic representable saddle functions in the definition environment for clarity.
- I found it weird to decouple the definition of saddle functions and saddle points, I would merge them
- Before section 2.4, when you mention differentiating through functions defined as a maximum, maybe you could quickly mention Danskin theorem, and or the implicit differentiation line of work
- I did not understand the "subgradient methods" paragraph in section 2.5
- In the example "robust cost LP" in section 2.6, maybe you could recall what is $\mathcal{C}$ to avoid the reader searching before

Clarity of Part 4:
- I am strongly against the notation $\phi(x, y)$ to denote a function, the standard notation is $\phi$, or $(x, y) \rightarrow \phi(x, y)$. I do not see the benefit of using $\phi(x, y)$. IMO the whole part 4 should be rewritten using $f$ for functions.
- **What is exactly a conically representable saddle point function? ** Is this a **bilinear constrained min max?** This is what I parse from the definition, but I am confused since non-bilinear examples are presented in the implementation section part 5.
What is t in the definition of conically representable saddle point functions? It seems it is not defined. Honestly I was very disappointed by this part. The authors did such a great pedagogical job in Parts 1 to 3, and dryly introduced their main concept, with a laconic  "Juditsky and Nemiroski 2022 for details"
- In section 4.2, at "automated dualization", what is the concept "power of the conic form"?
- In section 4.2, at "obtaining convex and concave saddle point coordinates", I was not able to understand why solving the problem with $f$ and $-f$ yields the saddle points from the solution of the convex problem

Typos:
- first equation in 2.5 ,= >> =

**Strengths And Weaknesses:**

Strength:
- Parts 1 to 3 are outstandingly clear, and filled with examples, the main idea, which is to dualize one of the optimization problems is clearly explained through an example. It provides a clear introduction to Juditsky and Nemiroski 2022.
- Code is great and modular

Weaknesses:
- For me, the main weakness of the paper is that Part 4, where the main contribution of the authors lies, is not clear enough.
In particular, I am not sure I correctly understood what type of problems the proposed framework is handling.
IMO this is  due to:
1 - The use of non-standard notation to define functions
2 - The definition of the main introduced object, "conically representable saddle functions", which should be much more polished
(see requested changes below)

---

> ### Author Response · Authors · 2023-09-08
> **Author Response to Reviewer s9NH**
>
> We thank the reviewer for their comments. We respond to the requested changes inline below.
>
> > Most of my concerns are regarding the clarity of Part 4
>
> > Contributions:
> I would merge the 2 first points in the contribution (p3, section 1.2)
>
> Thank you for the suggestion. We have merged the first two points.
>
> > Clarity before part 4:
> I found the definition of the mathematical objects very hard to parse, I had to read them multiple times to correctly understand what they were precisely. In particular, I would recommend encapsulating the mathematical definitions of saddle functions, saddle points, saddle problems, saddle extremum function, and conic representable saddle functions in the definition environment for clarity.
> I found it weird to decouple the definition of saddle functions and saddle points, I would merge them
> Before section 2.4, when you mention differentiating through functions defined as a maximum, maybe you could quickly mention Danskin theorem, and or the implicit differentiation line of work
> I did not understand the "subgradient methods" paragraph in section 2.5
> In the example "robust cost LP" in section 2.6, maybe you could recall what is
>  to avoid the reader searching before
>
> We feel is out of the scope of this paper to present
> subgradient methods, and as such we refer the reader to a relevant citation. We
> simply mention them here as they are an example of a specific method in the
> literature to solve saddle problems. As for the robust cost LP, we thank you for
> your suggestion and have expanded the inline expression to fully express the
> cost. Additionally, there is a hyperlink allowing the reader to jump to the
> relevant equation on the previous page.
>
> > I am strongly against the notation $\phi(x,y)$
>  to denote a function, the standard notation is $\phi$
> , or $(x,y)\to\phi(x,y)$. I do not see the benefit of using $\phi(x,y)$. IMO the
> whole part 4 should be rewritten using $f$ for functions.
>
> Thank you for your feedback. We have elected to retain our current notation
> since it matches that of the seminal Juditsky and Nemirovski paper.
>
> > **What is exactly a conically representable saddle point function? ** Is this a bilinear constrained min max? This is what I parse from the definition, but I am confused since non-bilinear examples are presented in the implementation section part 5. What is t in the definition of conically representable saddle point functions? It seems it is not defined. Honestly I was very disappointed by this part. The authors did such a great pedagogical job in Parts 1 to 3, and dryly introduced their main concept, with a laconic "Juditsky and Nemirovski 2022 for details"
>
> We apologize sincerely for the lack of clarity here. There was a missing
> subscript in the infimum that specified the inf was taken over the variables $f$, $t$, and $u$. We have fixed this error and added a line describing the
> dimensions of these objects. Our direction to Juditsky and Nemirovski was not
> intended to avoid proper definition; it was so the reader could get more
> background and technical conditions that are beyond the scope of this work.
>
> With respect to your question about bilinear functions: the conic form is a
> particular representation of a function, as you said as a bilinear constrained
> max at each $x$,$y$ pair. All the very much non-bilinear functions we use in DSP
> admit this representation. As is mentioned at the end of the section of the
> paper, this definition generalizes the class of bilinear saddle functions.
>
> > In section 4.2, at "automated dualization", what is the concept "power of the conic form"?
>
> Thank you for the comment, we have changed the ambiguous wording here to avoid confusion.
>
> > In section 4.2, at "obtaining convex and concave saddle point coordinates", I was not able to understand why solving the problem with
>  and
>  yields the saddle points from the solution of the convex problem
>
> We have rewritten the procedure for obtaining the convex and concave coordinates. We hope that the new version is clearer.
>
> > Typos:
> first equation in 2.5 ,= >> =
>
> Thank you, we have fixed the typo.

---

> > ### Comment · Reviewer_s9NH · 2023-09-08
> > **More question on Conically representable saddle functions**
> >
> > Just to be sure I correctly understood:
> > - The scope of the paper is Conically representable saddle functions, which are the functions which can be written as  $\inf f^T y + t| ....}$.
> > - If yes, is it obvious that this is a large class of functions, and can handle usual machine learning functions?
> > - Did I miss something about this in the paper?

---

> > > ### Author Response · Authors · 2023-09-08
> > > **Response to more questions**
> > >
> > > - To your first point, yes: the scope of DSP is saddle problems involving convex-concave saddle functions who admit the conic representation $\phi(x,y) = \inf_{f,t,u}\{f^Ty+t\mid Pf+tp+Qu+Rx\preceq_K s\}$.
> > > - This function class is large and includes all convex and concave functions. "Usual machine learning functions" do not usually have the form $g(x) = \sup_{y\in\mathcal{Y}} f(x,y)$. An application of DSP for machine learning is in robust model fitting, where the objective has the form $g(x) = \sup_{w\in\mathcal{W}}\sum_iw_if(x_i)$, which is handled by DSP for any convex loss $f$.
> > > - Sections 2.1 and 3.2 contain the above examples.

---

> > > > ### Author Response · Authors · 2024-01-18
> > > > **Summary of changes (part 1)**
> > > >
> > > > In addition to concerns raised by several reviewers addressed jointly in our [general comment](https://openreview.net/forum?id=KhMLfEIoUm&noteId=tbZLFJJkii), we respond point-to-point to individual comments below.
> > > >
> > > > **s9NH:**
> > > > Clarity of contributions:
> > > > I would merge the 2 first points in the contribution (p3, section 1.2)
> > > >
> > > > **Author response:** We have merged these two points, and modified the contribution paragraph in general in response to the other comments about technical contributions. We now state them as
> > > >
> > > > > - We introduce disciplined saddle programming, a domain specific language
> > > > for specifying and solving convex-concave saddle problems. To solve the saddle
> > > > problems, automated dualization is applied to the conic representation of the
> > > > problem. We extend the existing literature by deriving a procedure that
> > > > returns both the convex and concave coordinates of the saddle point.
> > > > This also guarantees that a valid saddle point was found without
> > > > the need to check for technical conditions (such as compactness).
> > > > These developments make the theory of conic representable saddle problems
> > > > practically applicable for the first time.
> > > > > - We specify and implement the first DSL that encodes sufficient conditions
> > > > for conic representability of saddle problems. We develop an open-source Python
> > > > package, also called DSP, providing a user-friendly interface for specifying and
> > > > solving saddle problems. Using this implementation, we demonstrate the
> > > > effectiveness of the framework by solving a variety of saddle problems from
> > > > different application domains.
> > > >
> > > > **s9NH:** Clarity before part 4:
> > > > I found the definition of the mathematical objects very hard to parse, I had to read them multiple times to correctly understand what they were precisely. In particular, I would recommend encapsulating the mathematical definitions of saddle functions, saddle points, saddle problems, saddle extremum function, and conic representable saddle functions in the definition environment for clarity.
> > > > I found it weird to decouple the definition of saddle functions and saddle points, I would merge them
> > > >
> > > > **Author response:** We decoupled the definitions of saddle functions and saddle points as the former also can appear in saddle extremum functions. We elected to keep the inline definitions over using a definition environment, as we perceived this as matter of style that was only mentioned by one reviewer.
> > > >
> > > > **s9NH:** Before section 2.4, when you mention differentiating through functions defined as a maximum, maybe you could quickly mention Danskin theorem, and or the implicit differentiation line of work. I did not understand the "subgradient methods" paragraph in section 2.5
> > > >
> > > > **Author response:** There was a grammatical error in the “subgradient methods” paragraph that is fixed in the latest revision. We also improved the wording of how such the subgradients of saddle functions can be found:
> > > > > We can readily compute a subgradient of a saddle max function (or a
> > > > supergradient of a saddle min function) at a given input, by simply maximizing
> > > > over the concave variable (minimizing over the convex variable), which is itself
> > > > a convex optimization problem, and then obtaining a subgradient (supergradient)
> > > > at that maximizer (minimizer).
> > > >
> > > > We could not find other reference to subgradients or differentiation, so we hope this clarifies the point. With respect to Danskin theorem or implicit differentiation, we do not think this is appropriate, since section 2.5 only relies on the simple properties of subgradients of a pointwise maximum.
> > > >
> > > > **s9NH:** In the example "robust cost LP" in section 2.6, maybe you could recall what is  to avoid the reader searching before
> > > >
> > > > **Author response:** We now recall the definition of C for clarity in the latest revision.

---

> > > > > ### Author Response · Authors · 2024-01-18
> > > > > **Summary of changes (part 2)**
> > > > >
> > > > > **s9NH:** Clarity of Part 4:
> > > > > I am strongly against the notation $\phi(x,y)$ to denote a function, the standard notation is $\phi$, or $(x,y)\to\phi(x,y)$. I do not see the benefit of using $\phi(x,y)$. IMO the whole part 4 should be rewritten using $f$ for functions.
> > > > > What is exactly a conically representable saddle point function? Is this a bilinear constrained min max? This is what I parse from the definition, but I am confused since non-bilinear examples are presented in the implementation section part 5. What is t in the definition of conically representable saddle point functions? It seems it is not defined. Honestly I was very disappointed by this part. The authors did such a great pedagogical job in Parts 1 to 3, and dryly introduced their main concept, with a laconic "Juditsky and Nemiroski 2022 for details"
> > > > > In section 4.2, at "automated dualization", what is the concept "power of the conic form"?
> > > > >
> > > > > **Author response:** We clarified numerous parts of our manuscript in response to this comment.
> > > > > First and foremost, we corrected our error about the previously undefined subscript, which made the description hard to parse, adding dimensions of the involved variables for improved clarity.
> > > > >
> > > > > We also expanded on the reasons behind the use of our notation, matching the preceding work by Judisky and Nemirovski and commented on the scope of conic representable saddle functions being broader than bilinear functions, with non-bilinear examples being given in the list of atoms in 5.1. Our direction to Juditsky and Nemirovski was not intended to avoid proper definition; it was so the reader could get more background and technical conditions that are beyond the scope of this work.
> > > > >
> > > > > Further, the sentence including the vague statement about the “power of the conic form” was replaced by “The conic form allows us to derive a dualized representation of the saddle extremum function $\Phi(x) = \sup_{y\in \mathcal{Y}}\phi(x,y)$ wich again admits a tractable conic form, meaning that it can be represented in a DSL like CVXPY.”
> > > > >
> > > > > **s9NH:** In section 4.2, at "obtaining convex and concave saddle point coordinates", I was not able to understand why solving the problem with  and  yields the saddle points from the solution of the convex problem
> > > > >
> > > > > **Author response:** We improved the clarity of the relevant paragraph in our initial response, and again improved it in the most recent revision. It now reads:
> > > > >
> > > > > > In our implementation, a saddle problem with an SE function in the objective
> > > > > is solved by applying the above automatic dualization to both the objective
> > > > > $\phi$ and $-\phi$ and then solving each resulting convex problem.
> > > > > Note that $\phi(x,y)$ is convex in $x$ and concave in $y$, while $-\phi(x,y)$ is
> > > > > concave in $x$ and convex in $y$. We do so in order to obtain both the convex and concave components of the saddle point, since the dualization removes the concave variable. To see this, note that equation 13 contains $x$ but not $y$ (and the opposite holds for the
> > > > > negated problem).

---

### Author Response · Authors · 2023-09-08
**Author Response to Reviews on Disciplined Saddle Programming**

We thank the reviewers for their feedback. We have addressed all their comments in the revised manuscript, which has substantially improved the paper. Below, we provide a detailed response to each point raised. We are particularly grateful that all five reviewers found the manuscript to be interesting to the audience of the journal, and that the claims made are supported by evidence. We are looking forward to the discussion phase, clarifying any remaining questions, and improving the manuscript further.

## Summary of changes:
- **Implementation** We have expanded the implementation section elaborating on how CVXPY is used to obtain the conic representation before the automated dualization takes place, which in turn yields a CVXPY problem. We address the questions about performance in detail below. We want to emphasize that like CVXPY, DSP acts as a *compiler* for saddle problems.
- **Theoretical contributions** We now highlight two important theoretical contributions of our paper, namely the procedure for obtaining the convex and concave coordinates of a saddle point problem, and the resulting guarantee that the saddle point problem was well posed without checking any of the sufficient conditions that are typically required to carry out the dualization by hand.
- Overall, we have improved the clarity of the paper by incorporating the reviewers' suggestions, improving the notation, and fixing typos.

### Performance comparison
Since the dualization happens during the compile phase of the program, which is typically much faster than the solve phase, we believe that in most cases the performance of the package is not a concern for most users. The actual solve time is not impacted by DSP. To address this comment, we have run a comparison of the manually implemented `sum_largest` atom in CVXPY and an equivalent DSP program. A chart comparing canonicalization and solve times between both approaches is linked below.
Even though the DSP framework is much more general compared to the few atoms that are manually implemented in CVXPY, the performance difference is insubstantial.
The below figure demonstrates that for both implementations, solve time (similar
since they solve equivalent problems)
dominates the compilation time.
We want to emphasize that for any deviation from the already implemented atoms in CVXPY, the state-of-the-art method is to dualize the problem by hand. For performance critical applications, DSP may be used to guide and validate an explicit derivation.

https://imageupload.io/ib/kryazW4mNMkILvO_1694150370.png

---

### Author Response · Authors · 2024-01-18
**Summary of revisions**

To improve the clarity of our initial response to both the requested and recommended changes, we provide a general response to points raised by multiple reviewers below, to which we refer in the updated individual reviewer responses.

**Contributions**
A recurring recommendation from several reviewers was to more distinctly differentiate our paper's contributions from existing literature, particularly referencing works by Juditsky and Nemirovski (toyQ, MvoZ, FqVv). Our key contributions include a novel procedure for solving saddle point problems in both convex and concave coordinates, and the validation of saddle points derived through this process, without the need for explicit verification of conditions like compactness. We have enhanced our paper's presentation in light of these inputs, adding limitations such as the possible need for extra atoms. Further, we've elaborated on the automation of the dualization process using conic duality (toyQ).
We also underscore that, aside from software engineering challenges in developing a DSL, our work encompasses substantial technical efforts like compiling a comprehensive list of conically representable saddle functions as initial DSL atoms, and formulating a parser logic to ensure the specification of a valid saddle problem, including the 'no mixing variables' rule.

**Performance**
On computational performance (toyQ, MvoZ, FqVv), we now explicitly mention that our method serves as a compiler, where compile time is relatively minor compared to solve time. Manual dualization of problems is typically more laborious and prone to errors. We compare the solve times of DSP-generated problems to manually dualized implementations, like `sum_largest` in CVXPY, presenting similar results. Although we provide a table for completeness, we stress that this comparison is limited as the duration of manual dualization is unquantifiable. Our focus with DSP is on streamlining this laborious process.

We present a table illustrating solve times for a basic `sum_largest` minimization problem across varying vector lengths `n`, noting identical objective values to high numerical precision.

| n      | dsp_time | cvxpy_time |
|--------|----------|------------|
| 100    | 0.002    | 0.002      |
| 1000   | 0.013    | 0.012      |
| 10000  | 0.133    | 0.126      |
| 100000 | 1.52     | 1.45       |

We have included a note in our revision to clarify that DSP functions as a compiler, not incorporating any optimization algorithms but utilizing solvers available through CVXPY.

**Implementation Challenges**
We have expanded our discussion on implementation challenges (toyQ, 7kLT, FqVv). For instance, in the "Obtaining convex and concave saddle point coordinates" paragraph, we address the complexities involved in translating conic representable saddle functions into a practical tool. This includes tackling issues like the automated removal of concave variables and adhering to technical conditions, which we believe should remain hidden from the user interface.

Further, in the "Solving a saddle point problem" paragraph, we detail the `solve()` method of a `SaddlePointProblem` object. This includes checks for DSP compliance, setting up an auxiliary problem, and dualization using CVXPY. The final solution provides values for both convex and concave variables. We emphasize DSP's role as a compiler, reliant on external solvers. We also discuss limitations such as the need for expanding the list of atoms and the disciplined approach required for function specification.

---

### Decision · Action_Editor_PRWh · 2023-11-03

**Recommendation:** Accept with minor revision

**Comment:**

This work provided an open source implementation of the methods in Juditsky and Nemiroski 2022, for solving certain saddle problems. One of the major contributions is the automation of the dualization process, making saddle problem formulation accessible to a broader audience who may not be familiar with the details of saddle point problems. The reviewers have generally praised the paper's clarity, usability, and practical relevance, highlighting its open source implementation.

Reviewers have also raised a few concerns. Most concerns received responses; e.g., a few reviewers asked about the technical contribution, and the authors answered in one response that the implementation challenges include "hiding compactness requirements from users, obtaining both
x and y portions of saddle points, and local variables".
Reviewers are mostly happy about the responses, but also highlight the needed changes, including "adding clarity of the novel section + explain in detail the implementation challenges".

Besides, there is a general issue that may need to be addressed. In the response to each reviewer, the authors seem to only reply to "Requested Changes" but not "concerns" or "weakness" points. Anyhow, the reviewers do not object too much on the paper. I think this is partially because some are answered in the response to all reviewers, and some of the concerns are answered in other places. For instance, a few reviewers asked about the challenges in implemention, but only the questions that appear in "Requested Changes" received direct response, and the concerns in other places "(ii) The technical contribution of the work" (Reviewer MvoZ's comment) did not received direct reponse. That being said, I think a brief response to major concerns of each reviewer is needed, with possibly a pointer to the comments in other places.

Summary: Before acceptance, the authors need to make the requested changes, especially those in the recommendation part of the reviewers, and let the reviewers know the changes.
I would also suggest to add responses to the major concerns of each reviewer, so that future readers do not get confused on whether the concerns are addressed or not.

**Audience:**

The paper is likely to be of interest to individuals in the field of optimization and machine learning, espcially those working on saddle point problems in machine learning. There are quite a few important applications in ML that can be cast as saddle point problems, so this paper is of interest to some audience in ML community.
There are concerns raised about the paper's relevance to a general machine learning audience. Expanding the relevance of the paper to a broader ML audience could enhance its appeal, but I think the current setting is interest to AT LEAST some inviduals in ML community.

**Claims And Evidence:**

The claims made in the submission are generally supported by evidence, including those on automated dualization, open source implementation and application examples.


Open-Source Implementation: The provision of an open-source implementation of DCP allows other researchers to use the method without knowing details of duality. The availability of code serves as strong supporting evidence.

Automated Dualization: The central contribution of automated dualization trick for saddle problems is well-documented. The authors provide clear explanations of the methodology and how it simplifies the formulation of saddle problems. The concept is supported by theoretical discussions and examples.

Applications: The paper demonstrates the practical relevance of saddle problems in various domains, providing concrete examples and use cases. This strengthens the claims about the broad applicability of the proposed methodology.

---

> ### Author Response · Authors · 2023-11-06
> **Author response to the comment on the decision**
>
> The authors would like to thank the action editor and the reviewers for the positive feedback and the conditional acceptance of our paper, which we greatly look forward to.
>
> We are glad to hear that the reviewers were mostly happy about our responses. In addition to our first set of edits with respect to novelty and implementation challenges, we have made an additional modification to our paper to improve these areas further.
>
> In our response to the reviewers we indeed grouped similar comments together, which might have cause the impression that we did not respond to all concerns raised in the reviews.
> Also, per the editors concern about weaknesses vs. recommended changes, we observed that many reviews naturally requested changes based on the weaknesses identified, in which case we addressed them jointly.
>
> The challenges in implementation were raised by reviewers 7kLT and toyQ, so we point to our response on this issue here for convenience:
> > We have modified the paper to emphasize several implementation challenges, such as hiding compactness requirements from users, obtaining both and portions of saddle points, and local variables.
>
> The changes made to the manuscript in response the challenges mentioned above also address a concern by reviewer MvoZ, who felt that we should highlight the technical contributions of the paper more clearly. We outline these technical insights which are required to realize the mathematic formulation provided by Juditsky and Nemirovski as a usable DSL.
>
> We hope this addresses the feedback about our response and we would kindly ask for advice if further to the manuscript are needed.